**Brief Communication**

# An improved pathway for autonomous bioluminescence imaging in eukaryotes

Ekaterina S. Shakhova [1,2,10], Tatiana A. Karataeva[1,2,10], Nadezhda M. Markina[1,2,10], Tatiana Mitiouchkina[1,2,10], Kseniia A. Palkina[1,2,10], Maxim M. Perfilov [1,2,10], Monika G. Wood [3], Trish T. Hoang[3], Mary P. Hall[3], Liliia I. Fakhranurova[1], Anna E. Alekberova[2], Alena K. Malyshevskaia[1,2], Dmitry A. Gorbachev[1,2], Evgenia N. Bugaeva[1], Ludmila K. Pletneva[1], Vladislav V. Babenko[4], Daria I. Boldyreva[4], Andrey Y. Gorokhovatsky[2], Anastasia V. Balakireva[1,2], Feng Gao[5,6], Vladimir V. Choob[1,7], Lance P. Encell[3], Keith V. Wood[8], Ilia V. Yampolsky [1,2,8,9], Karen S. Sarkisyan [1,2,5,6,8] ✉ & Alexander S. Mishin [1,2] ✉

The discovery of the bioluminescence pathway in the fungus *Neonothopanus nambi* enabled engineering of eukaryotes with self-sustained luminescence. However, the brightness of luminescence in heterologous hosts was limited by performance of the native fungal enzymes. Here we report optimized versions of the pathway that enhance bioluminescence by one to two orders of magnitude in plant, fungal and mammalian hosts, and enable longitudinal video-rate imaging.

The ability of organisms to develop luminescence relies on the biosynthesis of the light-emitting substrate, luciferin. For most bioluminescent species, which glow by oxidizing various luciferins, the full set of genes encoding the bioluminescence pathway is not understood. Only two pathways leading to luciferin biosynthesis are currently known: a branch of fatty acid metabolism from bacteria, encoded by the *lux* operon[1], and caffeic acid cycle—a branch of phenylpropanoid metabolism discovered in fungi[2]. The bacterial pathway has been known since the late 1980s; however, it was not widely applied in eukaryotes[3,4], probably due to low light output and toxicity of pathway intermediates[5]. In contrast, the discovery of enzymes catalyzing light-emitting caffeic acid cycle in the fungus *Neonothopanus nambi* (Fig. 1a) quickly translated into the development of multicellular organisms with autonomous luminescence[6–8] and reporter tools for transient expression assays in planta[9–11].

Similarly to other imaging tools sourced from nature[12], the wild-type fungal bioluminescence pathway (FBP1) performed suboptimally in heterologous hosts. Low enzymatic activity and limited stability of enzymes at physiologically relevant temperatures[2] resulted in modest light output even when expression was driven by strong viral promoters[6,7]: for example, in mammalian cells, substrate-free luminescence was only an order of magnitude stronger than the background noise, when detected with sensitive electron-multiplying charge-coupled device camera. In this Brief Communication, we aimed to improve the pathway to achieve robust luminescence across a range of heterologous hosts.

We applied directed evolution to luciferase nnLuz and hispidin-3-hydroxylase nnH3H from *N. nambi*. For nnLuz, we used consensus mutagenesis to identify three substitutions T99P, T192S and A199P, a combination of which resulted in nnLuz_v3 with increased stability and brighter luminescence in bacterial and mammalian cells (Supplementary Figs. 1–3). Further random mutagenesis of nnLuz_v3 led to the identification of nnLuz_v4 carrying four additional substitutions (I3S, N4T, F11L and I63T). nnLuz_v4 showed brighter expression-adjusted luminescence in bacteria and similar brightness in mammalian cells (Supplementary Fig. 4) and demonstrated improved thermostability (Supplementary Fig. 5) and catalytic activity (Supplementary Fig. 6) in yeast.

[1]Planta LLC, Moscow, Russia. [2]Shemyakin-Ovchinnikov Institute of Bioorganic Chemistry, Russian Academy of Sciences, Moscow, Russia. [3]Promega Corporation, Madison, WI, USA. [4]Lopukhin Federal Research and Clinical Center of Physical-Chemical Medicine of Federal Medical Biological Agency, Moscow, Russia. [5]Synthetic Biology Group, MRC Laboratory of Medical Sciences, London, UK. [6]Institute of Clinical Sciences, Faculty of Medicine and Imperial College Centre for Synthetic Biology, Imperial College London, London, UK. [7]Botanical Garden of Lomonosov Moscow State University, Moscow, Russia. [8]Light Bio Inc, Ketchum, ID, USA. [9]Pirogov Russian National Research Medical University, Moscow, Russia. [10]These authors contributed equally: Ekaterina S. Shakhova, Tatiana A. Karataeva, Nadezhda M. Markina, Tatiana Mitiouchkina, Kseniia A. Palkina, Maxim M. Perfilov. ✉e-mail: karen@light.bio; alexander@planta.bio

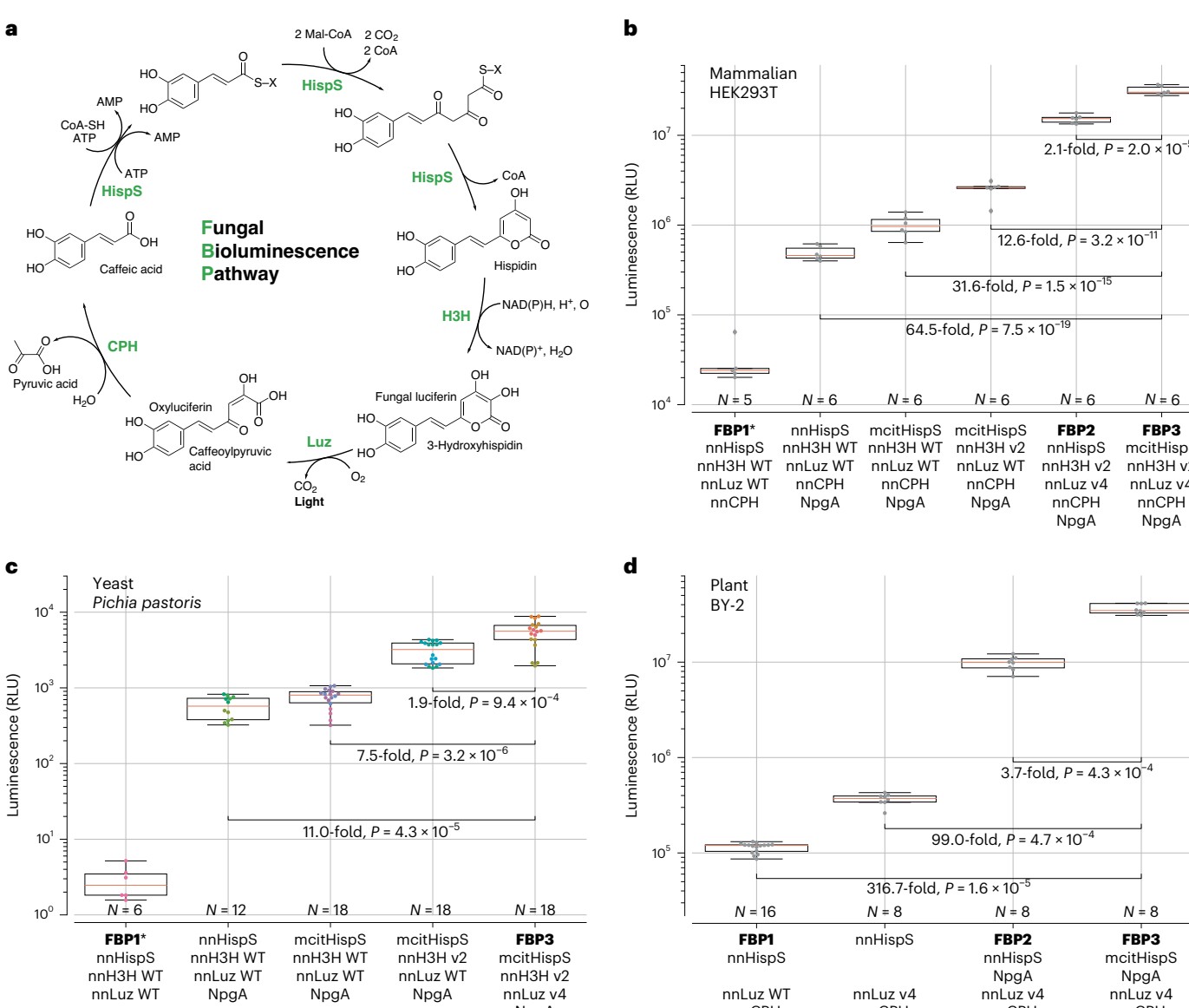

**Fig. 1 | Optimization of the fungal bioluminescence pathway in cell culture systems. a**, Biochemical reactions of the fungal bioluminescence pathway are catalyzed by hispidin synthase HispS, hispidin-3-hydroxylase H3H, luciferase Luz and putative caffeoyl pyruvate hydrolase CPH. Optimization of HispS, H3H and Luz-catalyzed steps resulted in two improved versions of the pathway: FBP2 and FBP3. **b**–**d**, FBP2 and FBP3 outperform the wild-type (WT) pathway when expressed in mammalian (**b**), yeast (**c**) and plant (**d**) hosts. In experiments in mammalian cells and yeast, each gene was delivered on a separate plasmid; in case of plants, plasmids encoding all genes were used. Experiments in plants and in yeast were performed at room temperature, and in mammalian cells at 37 °C. For mammalian cells, 100 μM caffeic acid was used, and for yeast 100 mM.

Comparison in yeast was performed in strains lacking nnCPH. Asterisk indicates samples where luminescence level was close to that of the background; for that reason, fold-change values are not provided. The boxes are the first and the third quartiles, whiskers are the rest of the distribution except outliers, and the orange line is the median. The color of data points in **c** indicates different yeast strains. The difference between mean values and $P$ values of post-hoc two-sided Conover test (**b**) or Mann−Whitney $U$ tests (**c** and **d**) corrected by the step-down method using Šidák adjustments are indicated below the brackets between the box plots. Kruskal−Wallis $H$ test: $H$-statistic 33.07, $P = 3.6 \times 10^{-6}$ (**b**), $H$-statistic 61.01, $P = 1.8 \times 10^{-12}$ (**c**), $H$-statistic 35.59, $P = 9.1 \times 10^{-8}$ (**d**). $N = 5−6$ biologically independent samples (**b**); 6−18 biologically independent samples (**c**); 8−16 plant cell packs (**d**).

Similarly, consensus mutagenesis of nnH3H identified five substitutions D37E, V181I, A183P, S323M and M385K that individually increased luminescence in mammalian cells (Supplementary Fig. 7). Screening of a combinatorial library of these mutations resulted in the identification of nnH3H_v2 (nnH3H D37E, V181I, S323M and M385K), which further enhanced the brightness of the pathway (Supplementary Figs. 8 and 9).

We then aimed to apply a similar strategy to the hispidin synthase nnHispS, but failed to detect any improved variants in our consensus mutagenesis library. We thus refocused our efforts on screening orthologous genes from other species of bioluminescent fungi and found

that hispidin synthase mcitHispS from *Mycena citricolor* significantly outperformed nnHispS in multiple heterologous hosts (*Nicotiana benthamiana* leaves, *Nicotiana tabacum* BY-2 cell culture, yeast *Pichia pastoris* and human HEK293T cells; Extended Data Fig. 1). Similarly to nnHispS, mcitHispS was efficiently activated by phosphopantetheinyl transferase NpgA from *Aspergillus nidulans*, which we confirmed to be a necessary component for bioluminescence in most tested plant species (Extended Data Fig. 2).

To assess joint performance of improved enzymes across yeast, plant and mammalian hosts, we combined them into two sets: FBP2

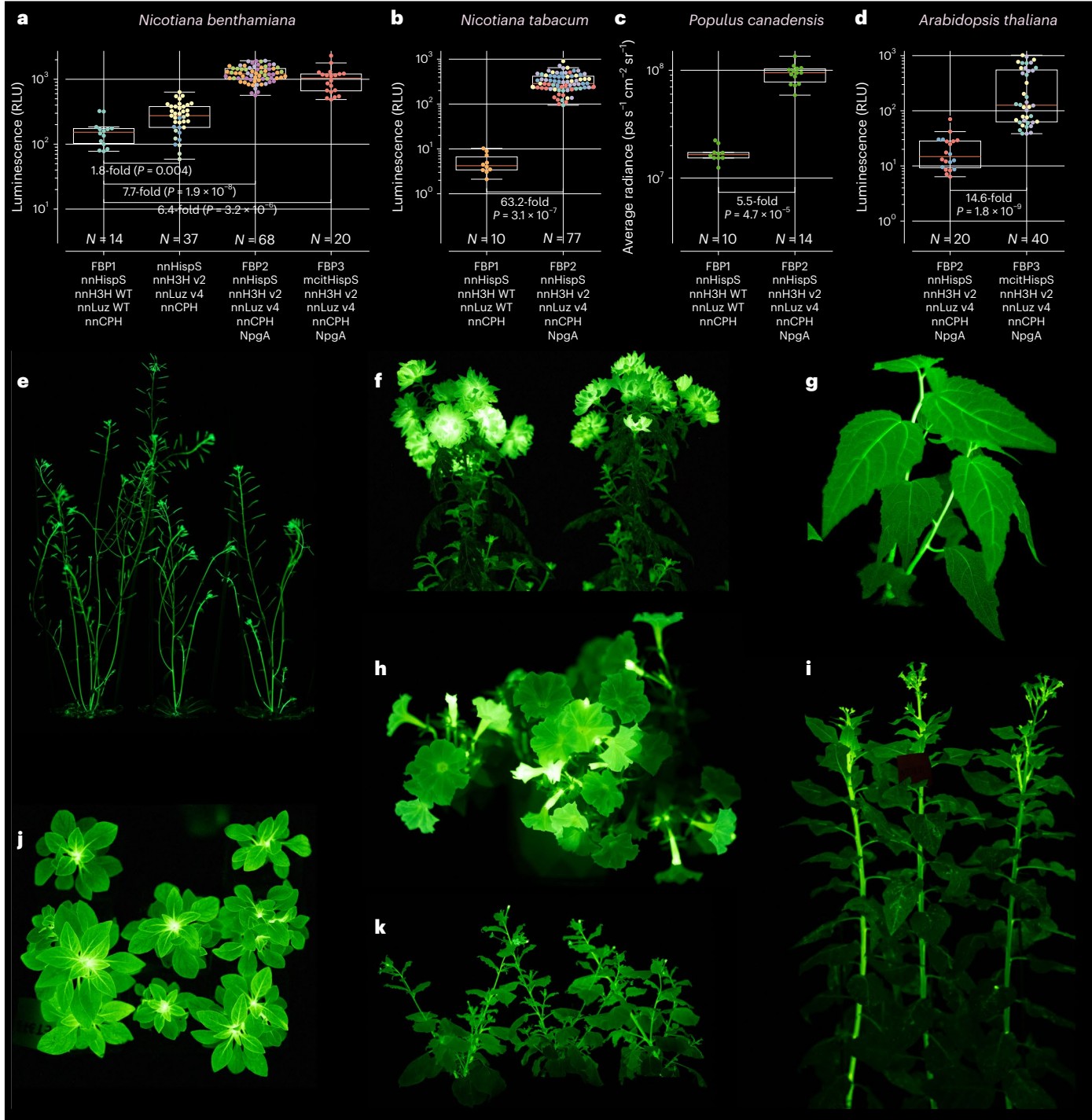

**Fig. 2 | Stable expression of pathway variants in plants. a–d**, Luminescence of transgenic lines stably expressing different versions of the bioluminescence pathway: average brightness of leaves of 3-week-old *Nicotiana benthamiana* (*N* = 14–68 leaves per box plot) (**a**), but also see Extended Data Fig. 5; average brightness of leaves of 4-week-old *Nicotiana tabacum* (*N* = 10–77 leaves per box plot) (**b**); average brightness of leaves of 3.5-month-old *Populus canadensis* (*N* = 10–14 leaves per box plot) (**c**); average brightness of leaves of 6-week-old *Arabidopsis thaliana* (*N* = 20–40 leaves per box plot) (**d**). The boxes are the first and the third quartiles, whiskers are the rest of the distribution except outliers, and the orange line is the median. The color of data points (if not neon green) indicates different plant lines. The difference between mean values and *P* values of post-hoc two-sided Mann–Whitney *U* tests corrected by the step-down method using Šidák adjustments are indicated below the brackets between the box plots. Photos of *Arabidopsis thaliana* (**e**), *Chrysanthemum morifolium* (**f**), *Populus canadensis* (**g**), *Petunia hybrida* (**h** and **j**), *Nicotiana tabacum* (**i**) and *Nicotiana benthamiana* (**k**) constitutively expressing FBP2, captured on Sony Alpha ILCE-7M3 camera (see Methods for ISO and exposure settings). Imaging was performed at room temperature. WT, wild type.

(nnHispS, nnH3H_v2, nnLuz_v4, nnCPH and NpgA) and FBP3 (mcitHispS, nnH3H_v2, nnLuz_v4, nnCPH and NpgA). In cell culture experiments, both sets of enzymes resulted in brighter luminescence, with

FBP3 outperforming FBP1 by one to two orders of magnitude across yeast, plant and mammalian hosts (Fig. 1b–d, Extended Data Fig. 3, Supplementary Figs. 10–13 and 26, and Supplementary Video 1).

When stably expressed from a genomic copy, the wild-type fungal pathway FBP1 performed well in tobacco species. However, in our hands its expression resulted in low or no light in other species. To assess whether FBP2 and FBP3 can broaden the applicability of self-sustained luminescence for plant biology, we created stable transgenic lines of six species representing four families of dicot plants. We chose plants that are used in diverse scientific and industrial contexts: model plants *Arabidopsis thaliana* and *Nicotiana benthamiana*, fast-growing tree *Populus canadensis*, ornamentals *Petunia hybrida* and *Chrysanthemum morifolium*, and industrially cultured tobacco *Nicotiana tabacum* (Fig. 2, Extended Data Figs. 4–6 and Supplementary Figs. 14–22). In contrast to FBP1, all plants expressing the assayed variants of the pathway were visibly glowing, with flowering petunia demonstrating the brightest bioluminescence visible to the naked eye without dark adaptation. Neither of the improved versions of the pathway led to noticeable adverse phenotypic changes in plants, or delayed yeast growth, compared to the wild-type pathway (Supplementary Figs. 23 and 24).

As many physiological events happen on short time scales, we assessed the ability of FBP2 and FBP3 to enable video-rate bioluminescence imaging in plants, using consumer-grade equipment. We showed that we could reliably monitor bioluminescence of petunia and tobacco plants using a consumer camera even in the presence of dim external lighting (Supplementary Video 2). Furthermore, the brightest tissues—petunia flower buds—could be recorded on modern smartphone cameras (Supplementary Video 3 and Supplementary Fig. 25).

Finally, we compared FBP3 to the optimized bacterial autoluminescence pathway iLux[4], and to NanoLuc and firefly luciferase—two commonly used luciferases that require exogenous substrate, in in vivo experiments. In plant cell culture, substrate-free luminescence of FBP3 was clearly visible in a dimly lit room, exceeding firefly luciferase by more than an order of magnitude, and approaching luminescence of NanoLuc supplied with exogenous substrate (Extended Data Fig. 7). In mammalian cells, caffeic-acid-induced luminescence of FBP3 was two orders of magnitude lower, compared to firefly luciferase, and three orders of magnitude lower, compared to NanoLuc (Extended Data Fig. 8). When compared to iLux in plant cells, substrate-free luminescence of FBP3 was two to five orders of magnitude brighter than that of the bacterial pathway, depending on gene dosage and subcellular localization of bacterial pathway components in the cell (Extended Data Fig. 9). In mammalian cells, caffeic-acid-induced luminescence of FBP3 was about fivefold dimmer than that of iLux (Extended Data Fig. 10).

Taken together, these results show that FBP2 and FBP3 confer robust bioluminescence in plants and fungi, with FBP3 demonstrating better performance than FBP2. Supply of exogenous caffeic acid is required for light emission in organisms that do not produce it biosynthetically, including mammalian cells and yeast *Pichia pastoris*. The performance of the pathway in animal cells remains suboptimal, highlighting the potential for further optimization.

In our pathway engineering efforts, co-expression with phosphopantetheinyl transferase had the strongest effect on light emission. Improvement depended on the host, possibly reflecting differences in endogenous phosphopantetheinyl transferase activity. For instance, expression of phosphopantetheinyl transferase was absolutely required for light emission in animal and yeast hosts. Contributions of other enzymes were host dependent too. In yeast, improved hispidin hydroxylase had the largest effect. In mammalian cells it was the luciferase. In plant cell culture, hispidin synthase and hispidin hydroxylase had comparable contributions.

Autoluminescence imaging enables data collection using inexpensive, consumer-grade equipment, and eliminates the need for purchasing the luciferin substrate. This makes organism-level longitudinal experiments both accessible and scalable. The enhanced brightness and robust performance of FBP2 and FBP3 across various plant species expand the range of experiments possible with this technology and allow for an order-of-magnitude higher temporal resolution. Additional improvements to the pathway are desirable to enhance performance in animal cells, allow viral delivery, and reduce the size and number of transcription units for greater user friendliness. But even at present, genetically encoded autoluminescence can be used to noninvasively monitor numerous physiologically relevant processes across an organism's lifespan.

## Online content

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

## Methods

### Consensus mutagenesis of nnLuz and nnH3H

pF4Ag shuttle vector (Promega) was used for gene expression in *Escherichia coli* (T7 promoter-driven expression) and mammalian cells (CMV promoter-driven expression). The pF4Ag vector backbone introduces a C-terminal 3x-FLAG-HiBiT tag[13,14]. Bacterial expression was performed in the *E. coli* strain KRX (Promega), which enables rhamnose-inducible expression of T7 RNA polymerase. Consensus variant plasmids were constructed using site-directed mutagenesis. To assess the performance of the mutants, expression was induced by 0.1% rhamnose. The cells were lysed by passive lysis buffer (Promega), incubated with 100 μM fungal luciferin (Fln), and the luminescence was measured in GloMax-Multi+ luminometer (Promega) at room temperature. Activity of HiBiT-tagged nnH3H variants was assayed in mammalian cells in co-transfection with nnLuz v3. As a proxy for specific activity, we used the ratio of luminescence (HiBiT/fungal signal) when treated with hispidin or Fln. Activity of each variant was normalized to the expression level of HiBiT, as quantified with HiBiT lytic reagent (Promega N3030).

### Random mutagenesis of nnLuz gene

The nnLuz libraries were prepared using Diversify PCR Random Mutagenesis Kit (630703 Takara) according to the manufacturer's instructions. A library containing on average 2.5 nucleotide changes per gene was used to transform chemically competent KRX *E. coli*. Five-thousand colonies were picked into individual wells of 96-well plates and grown at 37 °C. Gene expression was induced by 0.1% rhamnose. Assays were completed using a robotic screening platform that included lysis of the cells with passive lysis buffer (Promega), incubation with 100 μM Fln, and luminescence measurement with ClarioStar Plate reader (BMG). Clones that produced greater than 1.5-fold higher activity compared to the nnLuz v3 were assayed in a secondary screen following the same assay parameters as the primary screen. Clones that showed improved activity were sequenced and taken ahead for further validation and characterization.

### Activity of nnLuz and nnH3H mutants in mammalian cells

HEK293 cells (ATCC) were transfected using the FugeneHD transfection reagent (Promega) according to the manufacturer's instructions. For luminescence measurements, medium was removed and replaced with $CO_2$-independent medium (Life Technologies 18045), 10% fetal bovine serum. The substrate (hispidin or Fln) was added to cells to the final concentration 0.8 mM. The luminescence was measured in GloMax-Multi+ luminometer (Promega) at 37 °C. Activity of HiBiT-tagged nnLuz or nnH3H variants was normalized to the expression level of HiBiT, as quantified with HiBiT lytic reagent (Promega N3030).

### Enzyme stability assay in E. coli lysates

For enzyme stability measurements, gene expression in bacteria was performed as described above. To lyse the cells, we sonicated the suspensions in tris-buffered saline with addition of protease inhibitors cocktail (Promega G6521) and 0.2% dodecyl maltoside (DDM). Then samples were placed on an orbital mixer for 2 h at 4 °C. Cleared lysates were prepared by spinning cultures at 4,600*g* for 15 min. For the thermostability assay, we diluted the cleared lysates 1:100 into tris-buffered saline + 0.01% bovine serum albumin + 0.1% DDM. Samples were incubated at 22 °C, and at various time points aliquots were placed on ice (three technical replicates per sample). To assess the enzyme activity, we incubated the samples with 50 μM Fln. Samples were immediately placed in a GloMax-Multi+ luminometer (Promega) and read for signal output. In the GraphPad Prism one phase decay, least squares fit was used to calculate the half-life for each sample.

### Design and assembly of genetic constructs

Coding sequences of genes used in this study were optimized for the expression in *Nicotiana benthamiana*, *Pichia pastoris* or *Homo sapiens* and ordered synthetically (Supplementary Table 1). For genes encoding bacterial bioluminescence pathway (co-iLux), nucleotide sequences were obtained from[4]. Synthetic fragments were flanked by BpiI restriction sites designed to leave AATG–GCTT or AATG–AGGT overhangs, compatible with the modular Golden Gate-based assembly standard described in[15]. Gene of Firefly luciferase and NanoLuc were amplified from plasmid pGL3 (E1751, Promega) and pNL1.1 (N1001, Promega), respectively.

Golden Gate assembly was performed in the T4 ligase buffer (Thermo Fisher) containing 10 U of T4 ligase, 20 U of either BsaI or BpiI (Thermo Fisher) and ~100 ng of each DNA part. Typically, Golden Gate reactions were performed according to 'troubleshooting' cycling conditions described in ref. [16]: 25 cycles (90 s at 37 °C, 180 s at 16 °C), then 5 min at 50 °C and 10 min at 80 °C.

Correct DNA assembly was typically confirmed by Sanger sequencing, and in some cases additionally by Nanopore or Illumina-based whole plasmid sequencing. DNA assembly and whole-plasmid sequencing was typically ordered from Cloning Facility (cloning.tech).

Plasmids for expression in animal cells had the following insert structure: pCMV–gene–tSV40, and were assembled using DVK_AF vector as the backbone (CIDAR MoCLo Parts Kit # 1000000059; Supplementary Table 2).

Plasmids for expression in yeast were based on Golden Gate-compatible backbones with GAP promoter and AOX terminator driving expression of the inserted gene, and different selectable markers (kanamycin resistance, hygromycin resistance and zeocin resistance; Supplementary Table 3), with the exception of NpgA expression plasmid that had HIS4-based selection cassette and homology arms for OLE1 locus[17].

Plasmids for expression in plants. nnLuz variants were cloned into MoClo Level 1-like vector under the control of the 1.3 kb constitutive 35S promoter from cauliflower mosaic virus with 5′ untranslated region (UTR) of TMV omega virus and ocs terminator from *Agrobacterium tumefaciens*; HispS variants, nnCPH, FFLuc, NanoLuc were cloned into Level 1-like vector under the control of the 0.4 kb constitutive 35S promoter from cauliflower mosaic virus with 5′ UTR of TMV omega and ocs terminator from *Agrobacterium tumefaciens*; nnH3H WT/v2 gene was then cloned into Level 1-like vector, under the control of the constitutive FMV promoter from figwort mosaic virus and nopaline synthase terminator from *Agrobacterium tumefaciens*. *NpgA* gene was cloned into a Level 1-like vector, under the control of the constitutive CmYLCV 9.11 promoter from Cestrum yellow leaf curling virus and ATPase terminator from *Solanum lycopersicum*. Genes encoding autonomous bacterial luminescence were cloned in Level 1-like vector under the control of the 0.4 kb constitutive 35S promoter from cauliflower mosaic virus with 5′ UTR of TMV omega and ocs terminator from *Agrobacterium tumefaciens*. In the case of plastid localization, leader peptide from ref. [18] was used as an N-terminal tag.

These Level 1 plasmids were then digested by BpiI and assembled together into several Level M-like backbones, which then were digested by Bsa and assembled together into Level P-like vectors resulting in polycistronic cassette of the following order: xxHispS–NpgA–nnLuz WT/v4–nnCPH–nnH3H WT/v2. This gene cluster was preceded by a kanamycin resistance cassette for selection in plants. The entire construct, consisting of the kanamycin cassette plus luminescence genes, was flanked by *Agrobacterium tumefaciens* insertion sequences to facilitate *Agrobacterium*-mediated random integration of the construct into plant genomes (Extended Data Fig. 4). Plasmid encoding FBP1 (pX037; Addgene plasmid #167156) was obtained as described in ref. [7].

### Expression in mammalian cells and luminescence imaging

In different sets of experiments, HEK293NT (unknown origin) or HEK293 (ATCC) cell line was seeded in 96-well plates (ibidi, ibiTreat μ-Plate 96 Well Black). Cells were transfected with a mixture of the five plasmids that encoded *nnLuz*, *nnH3H*, *xxHispS*, *nnCPH* and *NpgA* by

PolyFect Transfection Reagent (QIAGEN) according to the manufacturer's protocol, using 2 µl PolyFect per well and a mixture of plasmids containing 44 ng of plasmid expressing *nnLuz*, and equivalent amounts of the other four plasmids according to their size. Transfected cells were grown in Dulbecco's modified Eagle medium (PanEco) supplemented with 10% fetal bovine serum (HyClone), 4 mM L-glutamine, 10 U ml$^{-1}$ penicillin and 10 µg ml$^{-1}$ streptomycin, at 37 °C and 5% $CO_2$. Twenty-four hours after transfection, the medium was changed to 25 µM or 100 µM caffeic acid solution (0.1% dimethyl sulfoxide (DMSO) in Dulbecco's phosphate-buffered saline (DPBS)) and luminescence was imaged for 23 min by IVIS Spectrum CT (PerkinElmer) at 37 °C every 9 s with open filter, binning of 16 and exposure of 5 s. The number of independent replicates is included in the figures' legends.

Image processing was performed using Living Image 4.5.5 software and custom Python scripts. For luminescence quantification, we used total flux (photons s$^{-1}$) after background subtraction. Background subtraction was performed using the following formula: signal = signal$_{raw}$ − (background$_{mean}$ − 3 × background$_{SD}$).

Comparison of FBP3 pathway with other luciferases in mammalian cells HEK293NT cells were seeded in 96-well plates (ibidi, ibiTreat µ-Plate 96 Well Black). For each well 2 µl of PolyFect Transfection reagent and 275 ng of plasmid pNK6265, pNK6272 (Supplementary Table 2) were used. Plasmids for the FBP3 pathway were obtained as described above. Twenty-four hours after transfection, the medium was changed to 100 µl of 100 µM caffeic acid solution (0.1% DMSO in DPBS)—for FBP3; minimum essential media (MEM) with 20 mM HEPES and 0.5 µl of substrate N113 (kit N1110, Promega)—for NanoLuc; 100 µM D-Ln (LUCK-100, GoldBio) in 100 mM Tris−HCl (pH 7.8) with 5 mM $MgCl_2$—for FFluc; Fln was dissolved in DMSO and added to 0.2 M sodium phosphate buffer (0.5 M $Na_2SO_4$, pH 8) to the final concentration of 100 µM. Plates were placed in Tecan Spark preheated at 37 °C. Luminescence was measured with an open filter and automatic attenuation with exposure of 1 s for 20 min. Data processing was performed using custom Python scripts. Integral signal was quantified by integration along the 'time' axis using the composite trapezoidal rule (trapz function from numpy Python package, v1.23.5).

### Comparison of FBP3 with the bacterial bioluminescence pathway (co-iLux) in mammalian cells

HEK293NT cells were seeded in 24-well plates (lumox multiwell, sarstedt). For each well, 4 µl of PolyFect Transfection reagent and 500 ng of plasmids were used; ratio of plasmids for co-iLux system was used as described in ref. 4, or FBP3 80 ng of plasmid expressing *nnLuz* was taken, and equivalent amounts of the other four plasmids according to their size. Twenty-four hours after transfection, the medium was changed to 300 µl of 100 µM caffeic acid solution (0.1% DMSO in MEM medium with 20 mM HEPES)—for FBP3; MEM medium with 20 mM HEPES—for co-iLux. Plates were placed in Tecan Spark preheated at 37 °C, luminescence was measured with an open filter and automatic attenuation with exposure of 1 s for 20 min. Data processing was performed using custom Python scripts. Integral signal was quantified by integration along the 'time' axis using the composite trapezoidal rule (trapz function from numpy Python package, v1.23.5).

### Expression in yeast and luminescence imaging

The protocol for cultivation and transformation of yeast *Pichia pastoris* GS115 was obtained from ref. 19. Integration of plasmids into the yeast genome was targeted at GAP promoter locus. Linearized plasmids were used for the transformation of electrically competent yeast cells. Colonies were selected using 200 mg ml$^{-1}$ of G418 or 200 mg ml$^{-1}$ of hygromycin or 50 mg ml$^{-1}$ of zeocin. Polymerase chain reaction (PCR)-based screening of yeast colonies was done by heating up colonies in 10 µl of 20 mM NaOH at 90 °C for 7 min and then using 1 µl of the resulting solution for direct PCR.

For HispS comparison experiments, we created a recipient strain by transforming the strain from ref. 2 that encoded pGAP−nnLuz_WT−tAOX1

(KanR) and pGAP−nnH3H_WT−tAOX1 (HIS4) in the genome with the plasmid pGAP−NpgA−tAOX1 (HygR). This strain was then further used for transformation of pGAP−xxHispS−tAOX1 (ZeoR).

For comparison of FBP versions, we created yeast strains by consecutive transformations with genome-integrating plasmids: pNK5696 [pGAP−NpgA−tAOX (HIS4)]; pNK5869/pNK3293/pNK5871/pNK5889/pNK5913/pNK3292/pNK3287/pNK5867/pNK3222/pNK3017/pNK3019 [pGAP−HispS variant−tAOX (ZeoR)], pNK5709/pNK5712 [pGAP−nnH3H variant−tAOX (HygR)], pNK5788/pNK5785 [pGAP−nnLuz variant−tAOX (KanR)].

For luminescence imaging, yeast culture grown on plate was resuspended in 50 µl of 1 M sorbitol. Five microliters of the suspension was then added in three replicates to yeast extract−peptone−dextrose (YPD) agar plates lacking antibiotics. Plates were incubated at 30 °C for 20−24 h, and then 5 µl of 100 µM, 100 mM or 220 mM caffeic acid solution in the DPBS buffer or 100 µM of hispidin solution in DPBS or 100 µM of luciferin solution was added to each yeast strain. Imaging was performed in Fusion Pulse (Vilber), with exposure of 0.1 or 5 s every 3 or 5 min for 1.0−1.5 h—for caffeic acid treatment, with exposure of 20 s for 20 min—for hispidin treatment, and with exposure of 0.5 s every 10 s for 10 min in case of luciferin treatment.

Processing of images was performed using FiJi ImageJ distribution (version 1.53t)[20] and custom Python scripts. For luminescence quantification mean values in the region of interest after background subtraction were used. Background subtraction was performed using the following formula: signal = signal$_{raw}$ − (background$_{mean}$ − 3 × background$_{SD}$).

For incubation of nnLuz WT or v4 in different temperatures, yeast strains expressing *nnLuz WT* or *v4* were grown on plates for 24 h, than resuspended in PBS to final OD$_{600}$ of 5, then 50 µl of yeast suspension was placed in each well of a 96-well PCR plate. Different parts of the plate were incubated for 10 min at different temperatures in gradient thermal cycler, then the plate was placed on ice for 5 min, and 10 µl of 1.2 mM luciferin solution were added to each well. Then 25 µl of suspension from each well was transferred to black 96-well plate with wells already containing 0.2 M sodium phosphate buffer with 0.5 M $Na_2SO_4$ and pH 8; the final concentration of luciferin was 50 µM. Imaging was performed in Fusion Pulse (Vilber), with exposure of 5 s every 15 s for 20 min.

Processing of images was performed using FiJi ImageJ distribution (version 1.53t)[20] and custom Python scripts. For luminescence quantification mean values in the region of interest after background subtraction were used. Background subtraction was performed using the following formula: signal = signal$_{raw}$ − (background$_{mean}$ − 3 × background$_{SD}$). Then values of each strain were normalized by the mean value for this strain at the lowest temperature.

For growth rate measurements, yeast strains were inoculated into 150 µl of YPD medium to a final optical density at 600 nm of 0.2−0.3 in 96-well plates. The plates were incubated at 30 °C with shaking at 500 rpm in Tecan Spark plate reader (Switzerland) for 16 h. Optical density at 600 nm was measured every 30 min.

### Activity of nnLuz wt and v4 with HiBiT-tag in yeast lysates

Yeast strains expressing nnLuz_wt/v4−HiBIT fusions were inoculated into YPD medium and cultured at 30 °C with 220 rpm for 18−20 h. Then 4 ml of each yeast suspension was centrifuged at 4 °C 5,000g, and resuspended in 400 µl of 100 mM MOPS (pH 7.5), 4 mM ethylenediaminetetraacetic acid, 2 mM tris(2-carboxyethyl)phosphine and 1 mM phenylmethylsulfonyl fluoride. After that, 50−100 µg of 200 µm zirconium beads (Ops Diagnostics) and one glass bead (*diameter* = 5 mm) were added to each sample. The samples were treated by bead mill homogenizer (TissueLyser LT, Qiagen) at 16,000g at 4 °C for 30 min with pauses, then samples were centrifuged for 15 min at 4 °C at 9,500g. Lysates were filtered through Spin-X centrifuge tube filter 0.22 µm (Costar) by centrifugation at 16,000g at 4 °C for 5 min. Then samples were diluted 25 times in the same buffer as used for lysis, and concentrated in Amicon Ultra-15 Centrifugal Filter Unit 10 kDa

(Millipore) to a volume of 200 µl. Then 3.5 µl of lysates was added to 72.5 µl of 0.2 M sodium phosphate buffer with 0.5 M $Na_2SO_4$ and pH 8 and transferred to black 96-well plate, 25 µl of Fln (4% DMSO, 0.4% DDM and 40 mM thioglycolic acid) were added to samples to final concentration 0.0244–50 µM. Luminescence was measured in Tecan Spark plate reader with exposure of 500 ms for 15 min. Activity of each variant was normalized to the expression level of HiBiT, as quantified with HiBiT lytic reagent (Promega N3030).

To quantify luminescence, intensity values of the wells were used, normalized by the integral values for HiBiT luminescence kinetics. Then, the integral signal for each sample was calculated. Fitting of the Michaelis–Menten model was performed with scipy package for Python (version 1.11.3), using the Michaelis–Menten equation: $v = V_{max} \times [S]/K_M + [S]$, where v is the velocity of the reaction, $V_{max}$ is the maximal rate of the reaction, [S] is the substrate concentration, and $K_M$ is the Michaelis–Menten constant.

### Transformation of *Agrobacterium tumefaciens*
Plasmids were transformed into competent cells of *Agrobacterium tumefaciens* AGL0 (ref. [21]), and clones were selected on LB (Luria-Bertani) agar plates containing 50 mg/L of rifampicin and an additional antibiotic, depending on the plasmid used for transformation (200 mg l$^{-1}$ of carbenicillin, 50 mg ml$^{-1}$ of kanamycin or 100 mg ml$^{-1}$ spectinomycin). Individual colonies were then inoculated into 10 ml of LB medium containing the same concentration of antibiotics. After overnight incubation at 28 °C with shaking at 220 rpm, cultures were centrifuged at 2,900g, resuspended in 25% glycerol and stored as glycerol stocks at −80 °C.

### *Nicotiana tabacum* BY-2 cell culture and luminescence imaging
BY-2 cell culture was grown in BY-2 medium (Murashige and Skoog (MS) with 0.2 mg l$^{-1}$ 2,4-dichlorophenoxyacetic acid, 200 mg l$^{-1}$ $KH_2PO_4$, 1 mg l$^{-1}$ thiamine, 100 mg l$^{-1}$ myo-inositol and 30 g l$^{-1}$ sucrose) at 27 °C by shaking at 130 rpm in darkness, with 2 ml of 1-week-old culture being transferred into new 200 ml of BY-2 medium every week[22].

Transformations of BY-2 cells were made according to a protocol adapted from ref. [23]. One-week-old BY-2 culture was pelleted in black 96-well plates to create cell packs and infiltrated by a mixture of several agrobacterial strains containing binary vectors. One of the strains encoded silencing inhibitor P19 (OD$_{600}$0.2), and others encoded bioluminescence genes (OD$_{600}$0.5). Plates were incubated at 80% humidity at 22 °C for 72 h before measurements of luminescence. Comparison of different sets of enzymes was done by co-infiltrating BY-2 cells with agrobacteria individually encoding bioluminescence enzymes.

Imaging of 96-well plates containing BY-2-based cell packs were made in a microplate reader Tecan Spark, luminescence was measured with an open filter and exposure of 1 s.

Processing of images was performed using custom Python scripts (Python version 3.10.12). For luminescence quantification values in the wells were used.

### Comparison of FBP3 with the bacterial bioluminescence pathway (co-iLux) in plant cells
To compare FBP3 and co-iLux in plant cells, we used mixtures of agrobacterial cultures, each transformed with a Level 1-like plasmid encoding genes of the corresponding bioluminescence pathway. We used an equal amount of each strain (determined by the optical density at 600 nm), to the final OD$_{600}$ of 0.6 in the case of infiltration in *Nicotiana benthamiana* leaves, and to OD$_{600}$ of 0.5 in the case of BY-2 cells.

### Comparison of FBP3 pathway with other luciferases in BY-2 cells
Transformations of BY-2 cells were made by agrobacterial strains encoding plasmids pNK3071 or pNK6260 or pNK6269 (Supplementary Table 4), according to the protocol described above. Forty-eight hours post-infiltration, BY-2 cells were supplemented with 150 µl of MS medium (M5524, Sigma-Aldrich; pH 5.7), containing 100 µM D-Ln (LUCK-100, GoldBio)−in the case of FFLuc, or 0.75 µl of substrate N113 (kit N1110, Promega) in the case of NanoLuc, or no substrate in the case of FBP3. Plates were imaged in Tecan Spark with an open filter and automatic attenuation at 0.1 s exposure times for 30 min. Data processing was performed using custom Python scripts. Integral signal was quantified by integration along the 'time' axis using the composite trapezoidal rule (trapz function from numpy Python package, v1.23.5).

### Agroinfiltration of *Nicotiana benthamiana* leaves and *Petunia hybrida* flowers
On the day before agroinfiltration, glycerol stocks of agrobacteria were inoculated into 10 ml of LB containing 100 µM of acetosyringone, 50 mg l$^{-1}$ of rifampicin and an additional antibiotic, depending on the plasmid which were used. The cultures were grown in the dark overnight at 28 °C with shaking at 220 rpm. The cultures were then centrifuged at 2,900g, resuspended in MMA buffer (10 mM 2-morpholinoethanesulphonic acid), 10 mM $MgCl_2$ and 200 µM acetosyringone), and incubated at 28 °C, 100 rpm for 3–4 h. Next, optical density at 600 nm was measured and used to dilute each culture to the optical density of 0.6. In addition, suspension of *Agrobacterium* containing a plasmid encoding pNOS-P19−tOCS was added at the optical density of 0.2. We then mixed agrobacterial strains to infiltrate leaves of 4–6-week-old *Nicotiana benthamiana* or petals of *Petunia hybrida*, using a 1-ml medical syringe without needle. At least three different plants were infiltrated for each experiment (the total number of flowers or leaves is indicated in the figure legends). Comparison of different HispS (Extended Data Fig. 1) was made by agroinfiltration of *Nicotiana benthamiana* leaves by combining of agrobacteria carrying nnLuz WT, nnH3H WT, NpgA, nnCPH and agrobacteria carrying xxHispS.

### Imaging of agroinfiltrated *Nicotiana benthamiana* leaves and *Petunia hybrida* flowers
Seventy-two hours after agroinfiltration *Nicotiana benthamiana* leaves and forty-eight hours after agroinfiltration *Petunia hybrida* flowers were detached and the luminescence was measured from the top side of each leaf by Sony Alpha ILCE-7M3 camera and 35-mm T1.5 ED AS UMC VDSLR lens (Samyang, -f/1.4) with an exposure of 5–30 s and ISO 400, 3,200 and 20,000.

Processing of images was performed using FiJi ImageJ distribution (version 1.53t)[20] and custom Python scripts. For luminescence quantification mean values in the region of interest after background subtraction were used. Background subtraction was performed using the following formula: signal = signal$_{raw}$ − background$_{mean}$.

### *Agrobacterium*-mediated transformation of *Nicotiana benthamiana* and *Nicotiana tabacum*
*Agrobacterium tumefaciens* strains AGL0 carrying plasmid pNK497 (for transformation of *Nicotiana tabacum*) and plasmid pNK511, pNK497 or pNK3071 (for transformation of *Nicotiana benthamiana*) were grown in flasks on a shaker overnight at 28 °C in LB medium supplemented with 25 mg l$^{-1}$ rifampicin and 50 mg l$^{-1}$ kanamycin. Bacterial cultures were diluted in liquid MS medium to an optical density of 0.6 at 600 nm. Leaf explants used for transformation experiments were cut from 2-week-old tobacco plants (*Nicotiana tabacum* cv. Petit Havana SR1, *Nicotiana benthamiana*) and incubated with bacterial culture for 20 min. Leaf explants were then placed onto filter paper overlaid on MS medium (MS salts, MS vitamin, 30 g l$^{-1}$ sucrose and 8 g l$^{-1}$ agar, pH 5.8) supplemented with 1 mg l$^{-1}$ 6-benzylaminopurine and 0.1 mg l$^{-1}$ indolyl acetic acid. Two days after inoculation, explants were transferred to the same medium supplemented with 500 mg l$^{-1}$ cefotaxime and 75 mg l$^{-1}$ kanamycin. Regeneration shoots were cut and grown on MS medium with antibiotics.

## *Agrobacterium*-mediated transformation of *Populus canadensis*

*Agrobacterium tumefaciens* strain AGL0 carrying plasmid pX037 or pNK497 was grown in flask on a shaker overnight at 28 °C in LB medium supplemented with 25 mg l⁻¹ rifampicin and 50 mg l⁻¹ kanamycin. Bacterial cultures were diluted in liquid MS medium to an optical density of 0.6 at 600 nm. Leaf explants used for transformation experiments were cut from 4-week-old poplar plants (*Populus × canadensis*) and incubated with bacterial culture for 20 min. Leaf explants were then placed onto filter paper overlaid on MS medium (MS salts, MS vitamin, 30 g l⁻¹ sucrose and 8 g l⁻¹ agar, pH 5.8) supplemented with 30 µg l⁻¹ 6-benzylaminopurine, 10 µg l⁻¹ indole-3-butyric acid and 0.8 µg l⁻¹ thidiazuron. Two days after inoculation, explants were transferred to the same medium supplemented with 400 mg l⁻¹ cefotaxime and 30 mg l⁻¹ kanamycin. Regeneration shoots were cut and grown on MS medium with antibiotics.

## Agrobacterium-mediated transformation of *Arabidopsis thaliana*

Transformation of *Arabidopsis thaliana* (Ecotype Columbia) was made according to the protocol taken from ref. 24 using *Agrobacterium tumefaciens* strain AGL0 carrying plasmid pNK497.

## Agrobacterium-mediated transformation of *Petunia hybrida*

*Agrobacterium tumefaciens* strains AGL0 carrying plasmid pNK497 or pNK3071 were grown in flasks on a shaker overnight at 28 °C in LB medium supplemented with 25 mg l⁻¹ rifampicin and 50 mg l⁻¹ kanamycin. Bacterial cultures were diluted in liquid MS medium to an optical density of 0.6 at 600 nm. Stem explants used for transformation experiments were cut from 3-week-old *Petunia hybrida* plants were firstly placed in MS with 1 mg l⁻¹ 6-benzylaminopurine, 1 mg l⁻¹ thidiazuron, 0.5 mg l⁻¹ zeatin and 0.5 mg l⁻¹ indole-3-acetic acid for 3 h for shoot initiation and incubated with bacterial culture for 20 min. Stem explants were then placed onto filter paper overlaid on MS medium (MS salts, MS vitamin, 30 g l⁻¹ sucrose and 8 g l⁻¹ agar, pH 5.8) supplemented with 2 mg l⁻¹ thidiazuron and 0.2 mg l⁻¹ indole-3-acetic acid. Two days after inoculation, explants were transferred to the same medium supplemented with 500 mg l⁻¹ cefotaxime and 30 mg l⁻¹ kanamycin. Regeneration shoots were cut and grown on MS medium with antibiotics.

## *Agrobacterium*-mediated transformation of *Chrysanthemum morifolium*

*Agrobacterium tumefaciens* strain AGL0 carrying plasmid pNK497 was grown in flask on a shaker overnight at 28 °C in LB medium supplemented with 25 mg l⁻¹ rifampicin and 50 mg l⁻¹ kanamycin. Bacterial cultures were diluted in liquid MS medium to an optical density of 0.6 at 600 nm. Leaf explants used for transformation experiments were cut from 4-week-old chrysanthemum plants (*Chrysanthemum morifolium* 'Snowdon White') were first placed in MS with 3 mg l⁻¹ 2,4-dichlorophenoxyacetic acid, 1 mg l⁻¹ 6-benzylaminopurine, 1 mg l⁻¹ kinetin and 5 mg l⁻¹ 6-(γ,γ-dimethylallylamino)purine for 3 h and then incubated with bacterial culture for 20 min. Leaf explants were then placed onto filter paper overlaid on MS medium (MS salts, Quoirin and Lepoivre medium, 30 g l⁻¹ sucrose and 8 g l⁻¹ agar, pH 5.8) supplemented with 200 µg l⁻¹ 6-benzylaminopurine and 50 µg l⁻¹ 1-naphthaleneacetic acid. Two days after inoculation, explants were transferred to the same medium supplemented with 500 mg l⁻¹ cefotaxime and 35 mg l⁻¹ kanamycin. Regeneration shoots were cut and grown on MS medium with antibiotics.

## Genotyping of transgenic plants

The Eppendorf tube with 100 mg of leaf material was placed in liquid nitrogen, and then material was homogenized with pestle. The genome DNA was extracted using the ExtractDNA Blood kit (Evrogen) according to the manufacturer's protocol, and direct PCR was performed on each inserted gene.

## Plant growth conditions

Plants were propagated on MS medium supplemented with 30 g l⁻¹ sucrose, 0.8 wt/vol agar (Panreac) and 0.3 mg l⁻¹ indole-3-butyric acid. In vitro cultures were incubated at 24 ± 1 °C with a 12–16-day photoperiod, with mixed cool white and red light (Cool White and Gro-Lux fluorescent lamps) at a light intensity of 40 µmol s⁻¹ m⁻². After root development, plantlets were transferred to 9-cm pots with sterilized soil (1:3 wt/wt mixture of sand and peat). Potted plants were placed in the greenhouse at 22 ± 2 °C under neutral day conditions (12 h light/12 h dark; 150 µmol s⁻¹ m⁻²) and 75% relative humidity.

## Plant imaging setup on consumer-grade photo cameras

Typically, T0 generation of plants was used for imaging, unless stated otherwise, with the exception except for *Arabidopsis thaliana*, in which case T2 generation was used. With the exception of iPhone-shot photos and videos, Sony Alpha ILCE-7M3 camera with a 35-mm T1.5 ED AS UMC VDSLR lens (Samyang, -f/1.4) was used to capture all photos presented in this paper. Depending on the experimental setup, lens aperture and other considerations, a range of ISO values from 400 to 20,000 was used, with exposure times from 5 s to 30 s. Most of the photos were captured with exposure time of 30 s and ISO 3,200. The following parameters were used to obtain photos for Fig. 2: Fig. 2e, *Arabidopsis thaliana* ISO 3,200, exp 30 s; Fig. 2f, *Chrysanthemum* sp. ISO 20,000, exp 30 s; Fig. 2g, *Populus canadensis* ISO 400, exp 10 min; Fig. 2h, *Petunia hybrida* ISO 6,400, exp 30 s; Fig. 2i, *Nicotiana tabacum* ISO 3,200, exp 30 s; Fig. 2j, *Petunia hybrida* ISO 20,000, exp 30 s; Fig. 2k, *Nicotiana benthamiana* ISO 3,200, exp 30 s.

The photos were then processed in the following way. First, a dark frame (raw photo obtained in the dark with the same settings) was per-channel subtracted from a raw photo of plants (LibRaw version 0.19.2, 4channels tool) to remove hot pixels and reduce noise. Optionally, an ImageJ plugin was applied[25] to remove outliers ('hot pixels'). For most photos, only the green channels (G and G2) were kept in the final image. Final images were rendered in pseudocolor with the 'GreenHot' linear lookup table from ImageJ (Supplementary Figs. 18, 21 and 23) or with the 'Inferno' symmetrical logarithmic colormap with linear threshold 10–300 from matplotlib package (version 3.7.1., Python version 3.10.12; Extended Data Fig. 6 and Supplementary Figs. 14, 19 and 20).

Processing of images was performed using FiJi ImageJ distribution (version 1.53t)[20] and custom Python scripts. For luminescence quantification of whole plant raw integral density (RawIntDen) in the region of interest, containing the whole plant, were taken. Background subtraction was performed using the following formula: $signal = RawIntDen - (area_{ROI} \times background_{mean}/area_{background})$.

## Plant imaging on IVIS Spectrum CT

Plant imaging on IVIS Spectrum CT was performed without filters in front of the camera, with exposure of 1 s and binning of 4×. The samples were acquired with 'D' settings of field of view. Ambient light image was taken after the luminescence measurements. Other settings were left at defaults.

Processing of images was performed using Living Image 4.5.5 software and custom Python scripts. For luminescence quantification average radiance (p s⁻¹ cm⁻² sr⁻¹) after background subtraction was used. Background subtraction was performed using the following formula: $signal = signal_{raw} - background_{mean}$.

## Data presentation and statistics

Most of the data are plotted as box plots implemented in the Seaborn (https://seaborn.pydata.org/, ver. 0.12.2) and matplotlib (https://matplotlib.org/, ver. 3.7.1) packages, using Python version 3.10.12. Unless noted otherwise in figure captions, the boxes on the graphs extend from the lower to upper quartile values of the data, the horizontal line represents the median, and whiskers represent the full data range except outliers. Gray or colored dots represent individual values. Pairwise

post-hoc two-sided Mann–Whitney $U$ tests (scikit-posthocs package[26], version 0.8.0) with $P$ values corrected by the step-down method using Šidák adjustments were computed (Figs. 1c,d and 2, Extended Data Figs. 1, 2, 5b, 6, 7, 8 and 10, Supplementary Figs. 2, 9, 10c,d, 11–14, 17, 19 and 20). Kruskal–Wallis $H$ tests (scipy.stats package, https://www.scipy.org/, SciPy version 1.11.3) followed ($H_0$ was rejected) by multiple pairwise post-hoc two-sided Conover's tests (scikit-posthocs package[26], version 0.8.0) with $P$ values corrected by the step-down method using Šidák adjustments were computed (Fig. 1b, Extended Data Figs. 3, 5a and 9, and Supplementary Figs. 1, 4, 7, 10a,b, 15 and 16). Sample numbers ($N$) are reported in the figure or figure legend.

### Reporting summary

Further information on research design is available in the Nature Portfolio Reporting Summary linked to this article.

## Data availability

The plasmids used in this study will be made available for noncommercial use through Addgene. Data are available at https://doi.org/10.6084/m9.figshare.24623817.

## Code availability

Python code for processing and plotting data is available at both https://github.com/Perfus/BL2.0 and https://doi.org/10.6084/m9.figshare.24623976.

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

## Acknowledgements

This study was partially funded by Light Bio (light.bio) and Planta (planta.bio). We thank Milaboratory (milaboratory.com) for the access to computing and storage infrastructure. The Synthetic Biology Group is funded by the MRC Laboratory of Medical Sciences (UKRI MC-A658-5QEA0). We thank J. Petrasek for providing the BY-2 cell culture line. Whole-plant imaging experiments were funded by RFBR and MOST, project number 21-54-52004. Plant transformations were funded by RFBR and GACR, project number 20-54-26009. Luminescent assays in yeast were funded by RSF, project number 22-44-02024 (https://rscf.ru/project/22-44-02024/). Luminescence assays in BY-2 were funded by RSF, project number 22-74-00124 (https://rscf.ru/project/22-74-00124/). DNA assembly and agrobacterium-based assays were funded by RSF, project number 22-14-00400 (https://rscf.ru/project/22-14-00400/). We thank A. Kraevsky, S. Kostromin and I. Ivashkin for preparation of Supplementary Video 2. We thank Konstantin Lukyanov lab, and Sergey Deyev lab for assistance with experiments.

## Author contributions

E.S.S., T.A.K., N.M.M., T.M., K.A.P., M.M.P., M.G.W., T.T.H., M.P.H., L.I.F., A.E.B., A.K.M., D.A.G., E.N.B., L.K.P., V.V.B., D.I.B., F.G. and A.S.M. performed experiments. E.S.S., T.A.K., N.M.M., T.M., K.A.P., M.M.P., M.G.W., T.T.H., M.P.H., L.I.F., A.E.B., A.K.M., D.A.G., E.N.B., A.Y.G., L.K.P., A.Y.G., V.V.C., L.P.E., I.V.Y., K.V.W., K.S.S. and A.S.M. planned experimentation. M.M.P., E.S.S., K.S.S. and A.S.M. performed data analysis. E.S.S., M.M.P., A.V.B., K.S.S. and A.S.M. wrote the paper. I.V.Y., K.V.W., K.S.S. and A.S.M. proposed and directed the study. All authors reviewed the paper draft. The authors wish it to be known that, in their opinion, six first authors should be considered as joint first authors.

## Competing interests

This study was partially funded by Light Bio (light.bio) and Planta (planta.bio). K.V.W. is the CEO of Light Bio. K.S.S., K.V.W. and I.V.Y. are shareholders of Light Bio. A.S.M. is the CSO of Planta. E.S.S., T.A.K., N.M.M., T.M., K.A.P., M.M.P., L.I.F., A.K.M., E.N.B., L.K.P., A.V.B. and V.V.C. are employees of Planta. A.E.B., D.A.G., V.V.B., D.I.B., A.Y.G., F.G., M.G.W., T.T.H., M.P.H. and L.P.E. declare no competing interests.

## Additional information

**Extended data** is available for this paper at https://doi.org/10.1038/s41592-023-02152-y.

**Correspondence and requests for materials** should be addressed to Karen S. Sarkisyan or Alexander S. Mishin.

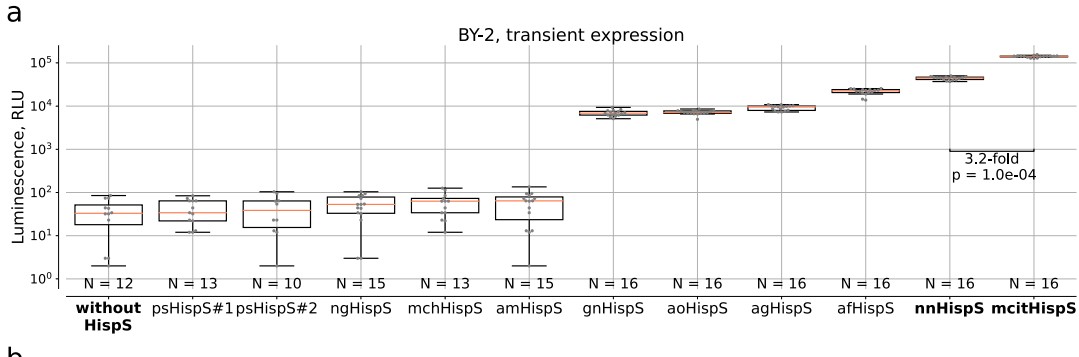

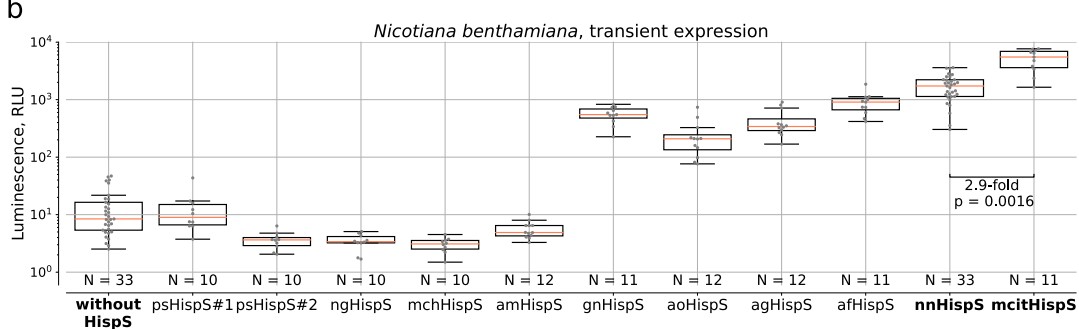

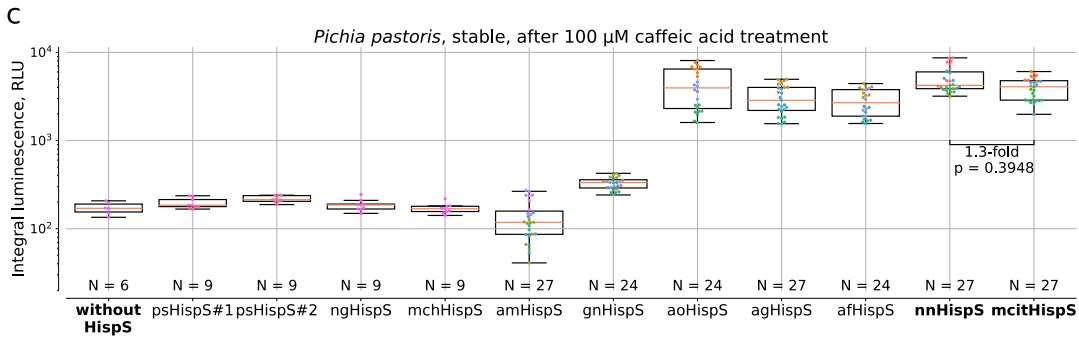

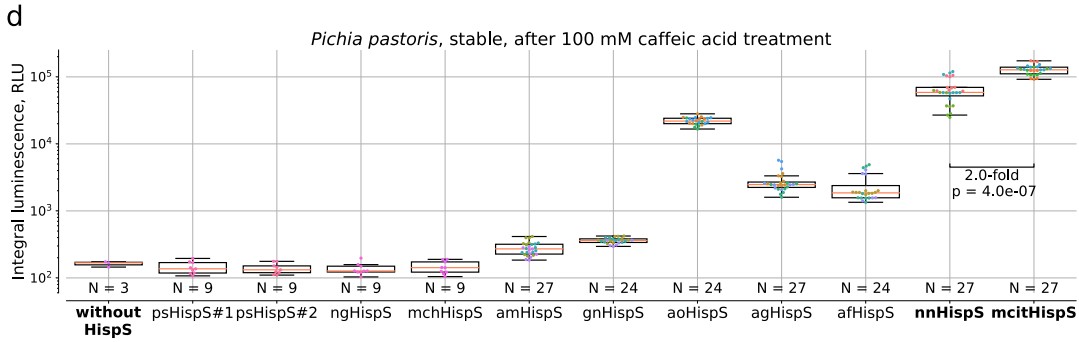

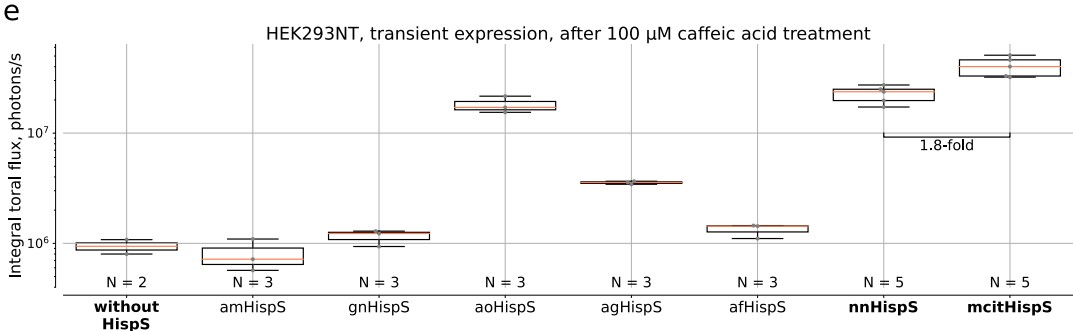

**Extended Data Fig. 1 | See next page for caption.**

**Extended Data Fig. 1 | Co-expression of HispS homologues with wild-type nnLuz and wild-type nnH3H in plant, yeast, and mammalian hosts.**
**a**. Transient expression in BY-2 plant cell packs (N = 12–16 cell packs per box plot).
**b**. Transient expression in *N. benthamiana* leaves (N = 10–33 leaves per box plot).
**c, d**. Expression from genome-integrated copy in yeast *P. pastoris*, luminescence assayed after adding 100 µM (c) and 100 mM (d) caffeic acid, integral signal for 1 hour (N = 6–27 and 3–27 biologically independent samples per box plot in c and d, respectively). **e**. Transient expression in human cell culture HEK293NT, luminescence assayed after adding 100 µM caffeic acid, integral signal for 23 mins (N = 2–5 biologically independent samples per box plot). The boxes are the first and the third quartiles, whiskers are the rest of the distribution except outliers, the orange line is the median. The colour of data points in case of c. and d. indicates different yeast strains (1–9 strains per box plot). The difference between mean values of nnHispS and mcitHispS and p-values of pairwise post-hoc two-sided Mann-Whitney U-tests (if applicable) corrected by the step-down method using Sidak adjustments are indicated below the brackets between the box plots. Kruskal-Wallis H Test: H-statistic = 155.79, p = 9.7e-28 (a), H-statistic = 158.86, p = 2.3e-28 (b), H-statistic = 182.56, p = 3.0e-33 (c), H-statistic = 24.79, p = 8.3e-04 (d), H-statistic = 155.79, p = 9.7e-28 (e). In these experiments, each gene was delivered on a separate plasmid. Plasmids used in these experiments are listed in Supplementary Table 1.

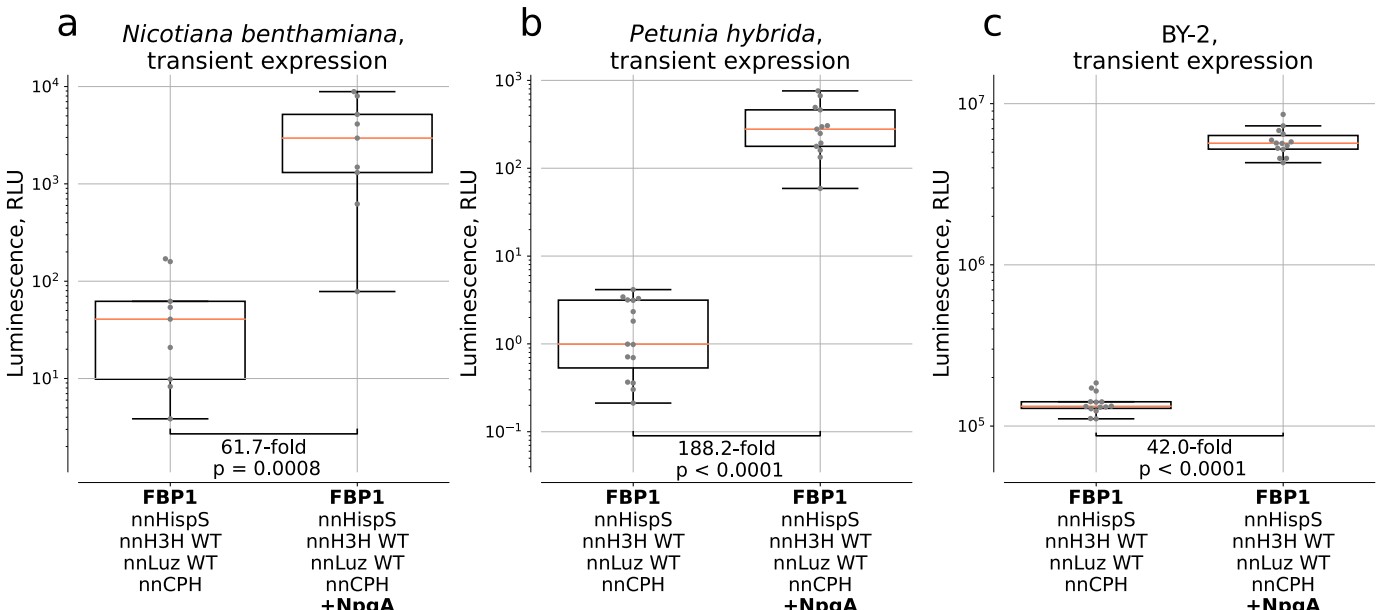

**Extended Data Fig. 2 | Effect of phosphopantetheinyl transferase NpgA on luminescence in plant cells. a**. Transient expression in *N. benthamiana* leaves (N = 9 leaves per box plot). **b**. Transient expression in *P. hybrida* flowers (N = 13 and N = 15 leaves for FBP1 + NpgA and FBP1, respectively). **c**. Transient expression in BY-2 cells (N = 14 plant cell packs per box plot). In these experiments, pX037 plasmid was co-infiltrated with a plasmid encoding NpgA. The boxes are the first and the third quartiles, whiskers are the rest of the distribution except outliers, the orange line is the median. The difference between mean values and p-values of post-hoc two-sided Mann-Whitney U-tests are indicated below the brackets between the box plots, p = 7.9e-04 (a), p = 8.0e-06 (b), p = 7.0e-06 (c). In these experiments, each gene was delivered on a separate plasmid.

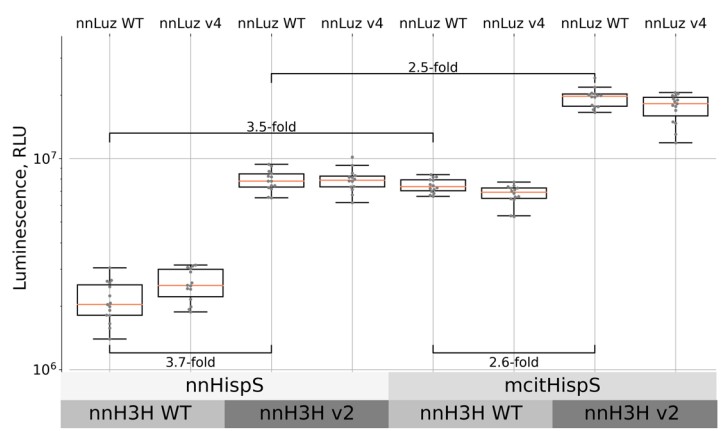

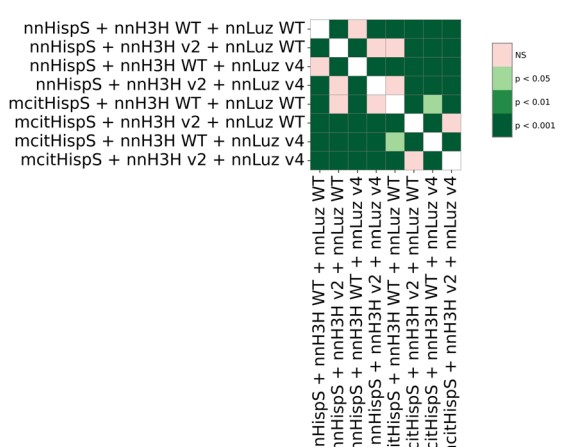

**Extended Data Fig. 3 | Comparison of combinations of nnLuz, nnH3H and HispS variants upon transient expression in BY-2 cells, upon co-expression with NpgA.** In these experiments, each gene was delivered on a separate plasmid. Box and whisker plots (left) are accompanied by colour-coded p-values of post-hoc two-sided Conover's test corrected by the step-down method using Sidak adjustments (right). NS − non-significant. The boxes are the first and the third quartiles, whiskers are the rest of the distribution except outliers, the orange line is the median. N = 15 plant cell packs for nnHispS + nnH3H WT + nnLuz WT and mcitHispS + nnH3H v2 + nnLuz v4 or 14 plant cell packs for other combinations. Kruskal-Wallis H Test: H-statistic = 99.89, p = 1.1e-18. The difference between several mean values are indicated below the brackets between the box plots.

### FBP1 (Mitiouchkina et al. 2020)

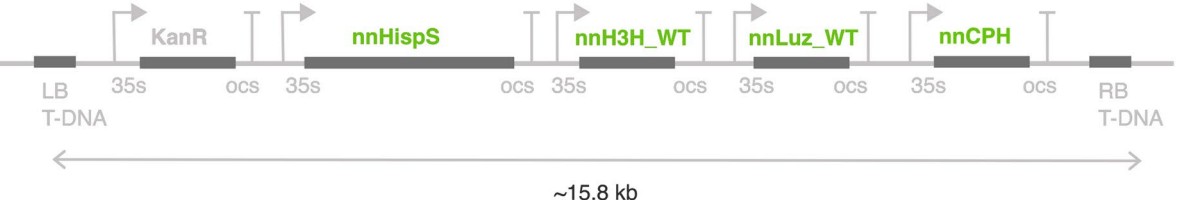

### nnHispS nnH3H_v2 nnLuz_v4 nnCPH

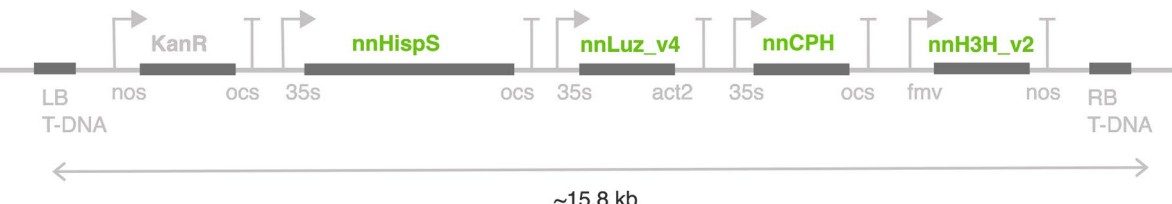

### FBP2

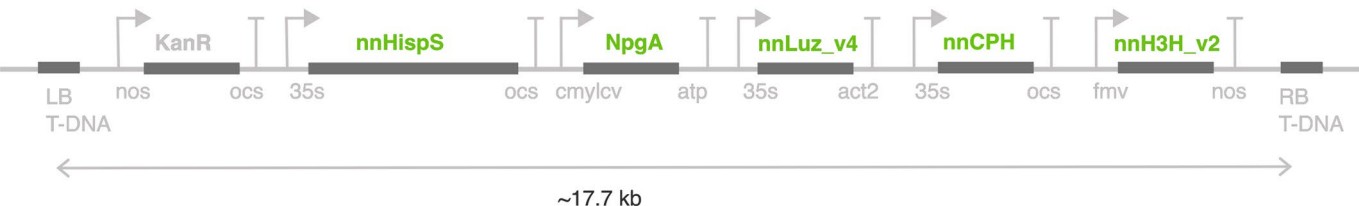

### FBP3

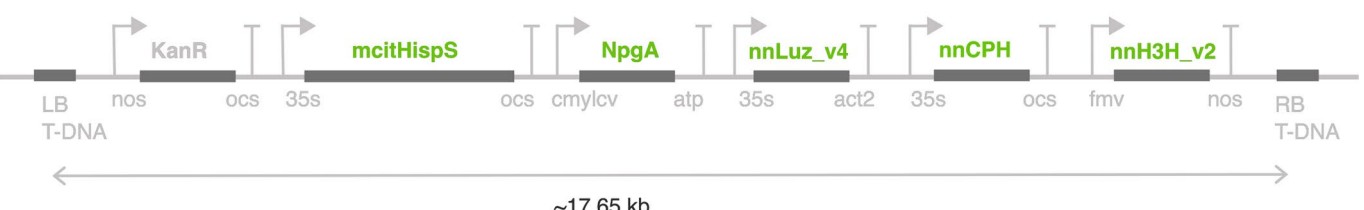

**Extended Data Fig. 4 | Insert structure of the plasmids used to create transgenic plant lines.**

*Nicotiana benthamiana*, transgenic lines

**Extended Data Fig. 5 | Average luminescence of leaves of 3-week-old *N. benthamiana* transformed with various versions of the pathway. a.** Same as Fig. 2a, provided here for easier comparison with (b): results of an experiment, which only had a single FBP3 transgenic line in comparison, for technical reasons. **b.** Results of an independent experiment, which only had a single FBP2 transgenic line in comparison, for technical reasons. The FBP1 is the plant line NB021 reported in ref. 7. Photos were captured with ISO 400 and 30 seconds of exposure (see Methods). Box and whisker plots are accompanied by colour-coded p-values of post-hoc two-sided Conover's test corrected by the step-down method using

Sidak adjustments. NS − non-significant. The boxes are the first and the third quartiles, whiskers are the rest of the distribution except outliers, the orange line is the median. The colour of data points indicates different plant lines: for a. NB021 for FBP1 ($N = 14$ leaves), 3 lines for nnHispS + nnH3H v2 + nnLuz v4 + nnCPH ($N = 37$ leaves), 4 for FBP2 ($N = 68$ leaves) and 1 for FBP3 ($N = 20$ leaves). b. One line for FBP2 ($N = 11$ leaves) and 4 lines for FBP3 ($N = 41$ leaves). The difference between mean values is indicated below the brackets between the box plots; in (b) supplied with p-value of post-hoc two-sided Mann-Whitney U-test, $p = 7.0e{-}06$. Kruskal-Wallis H Test: H-statistic = 98.97, $p = 2.6e{-}2$ (a).

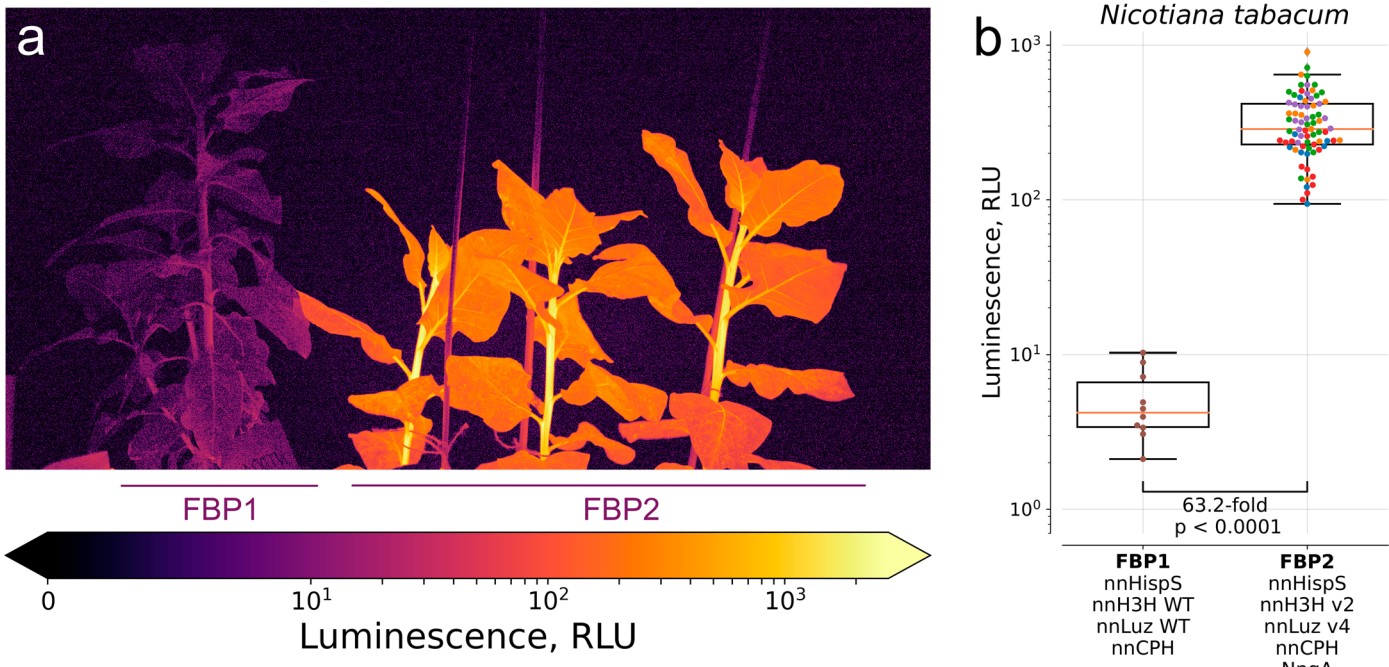

**Extended Data Fig. 6 | Imaging of 4-week-old *N. tabacum* transformed with FBP1 or FBP2.** The FBP1 is the plant line NT001 reported in ref. 7. The photo (ISO 400, exposure 30 sec) (**a**) and average luminescence of leaves (**b**). The boxes are the first and the third quartiles, whiskers are the rest of the distribution except outliers, the orange line is the median. The colour of data points indicates different plant lines (NT001 for FBP1, 5 lines for FBP2); N = 10 and N = 77 leaves for FBP1 and FBP2, respectively. The difference between mean values and p-value of post-hoc two-sided Mann-Whitney U-test are indicated below the brackets between the box plots, p = 3.1e-07.

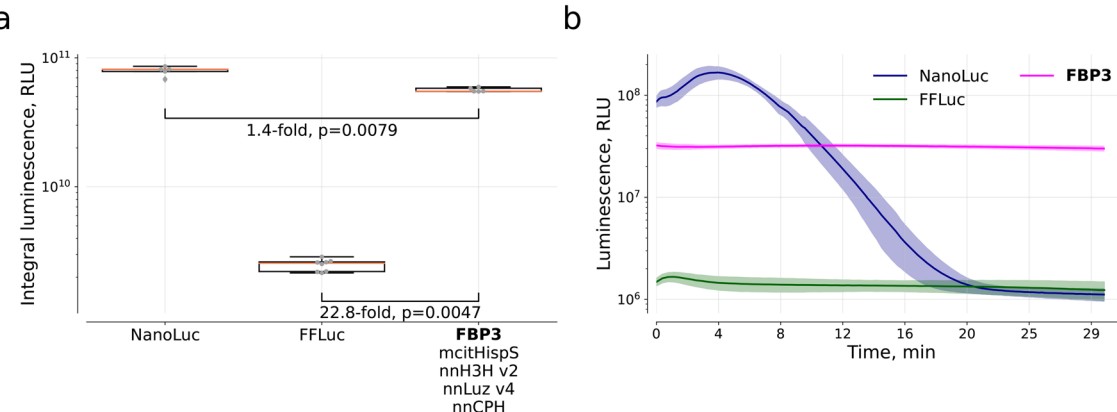

**Extended Data Fig. 7 | Comparison of FBP3 pathway to luciferases that require exogenous substrate in BY-2 cells. (a)** Integral luminescence signal collected for 30 minutes. The boxes are the first and the third quartiles, whiskers are the rest of the distribution except outliers, the orange line is the median. N = 5 plant cell packs for NanoLuc and FBP3, and N = 8 for FFLuc. The difference between mean values and p-values of post-hoc two-sided Mann-Whitney U-test corrected by the step-down method using Sidak adjustments are indicated below the brackets between the box plots. **(b)** - Kinetics. Data shown as mean (solid line) ± SD (area around the solid line). For NanoLuc, 0.75 μL of substrate N113 (N1110, Promega) was used. For FFLuc, we used 100 μM of D-Ln. For FBP3, no substrate was added.

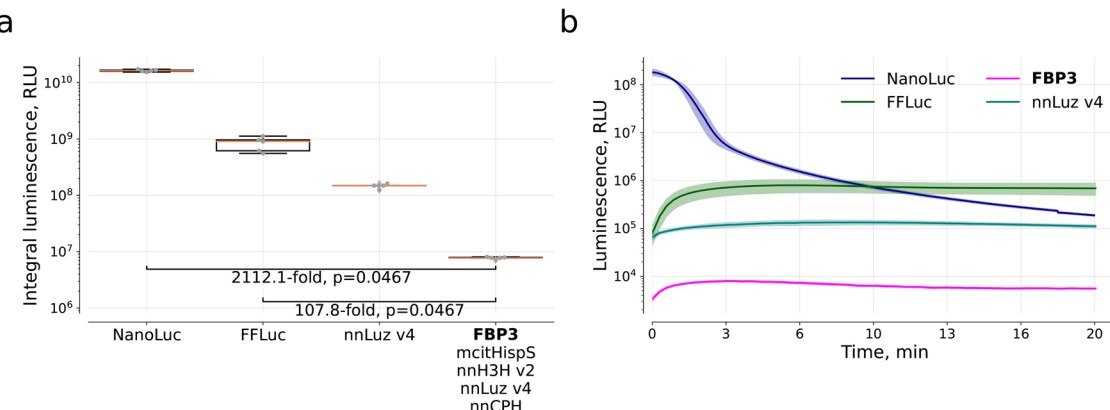

**Extended Data Fig. 8 | Comparison of FBP3 pathway to luciferases that require exogenous substrate in HEK293NT cells. (a)** Integral luminescence signal collected over 20 minutes. The boxes are the first and the third quartiles, whiskers are the rest of the distribution except outliers, the orange line is the median. N = 5 biologically independent samples per box plot. The difference between mean values and p-values of post-hoc two-sided Mann-Whitney U-test corrected by the step-down method using Sidak adjustments are indicated below the brackets between the box plots. **(b)** Kinetics. Data shown as mean (solid line) ± SD (area around the solid line). For NanoLuc, 0.5 μL of substrate N113 (N1110, Promega) was used. For FFLuc, we used 100 μM of D-Ln. For nnLuz v4, 100 μM of fungal luciferin was added. For FBP3, 100 μM of caffeic acid.

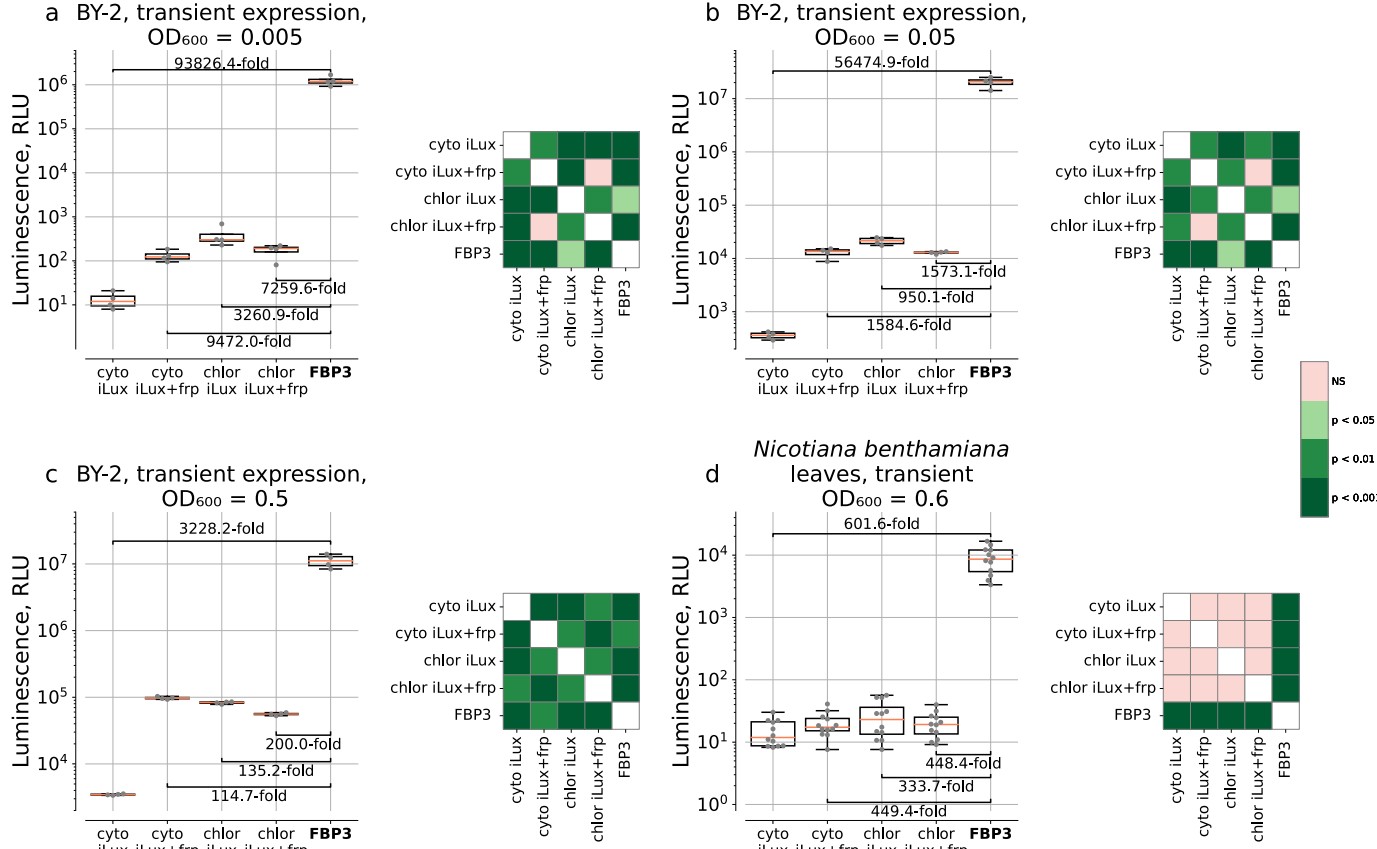

**Extended Data Fig. 9 | Autonomous luminescence conferred by bacterial and fungal pathways in plant cells.** (**a–c**) Luminescence of BY-2 cells infected by different amounts of agrobacteria, expressed as $OD_{600}$. (**d**) Luminescence of *N. benthamiana* leaves. In these experiments, each gene was delivered on a separate plasmid. *cyto* stands for cytoplasmic localisation of iLux enzymes, *chlor* – for plastid localisation. Box and whisker plots (left) are accompanied by colour-coded p-values of post-hoc two-sided Conover's test (right) corrected by the step-down method using Sidak adjustments. NS − non-significant. The boxes are the first and the third quartiles, whiskers are the rest of the distribution except outliers, the orange line is the median. N = 4 plant cell packs per box plot (a, b, c) or 12 leaves per box plot (d). The difference between mean values is indicated below the brackets between the box plots. Kruskal-Wallis H Test: H-statistic = 17.60, p = 1.5e-03 (a), H-statistic = 17.39, p = 1.6e-03 (b), H-statistic = 18.29, p = 1.1e-03 (c) and H-statistic = 30.86, p = 3.3e-06 (d).

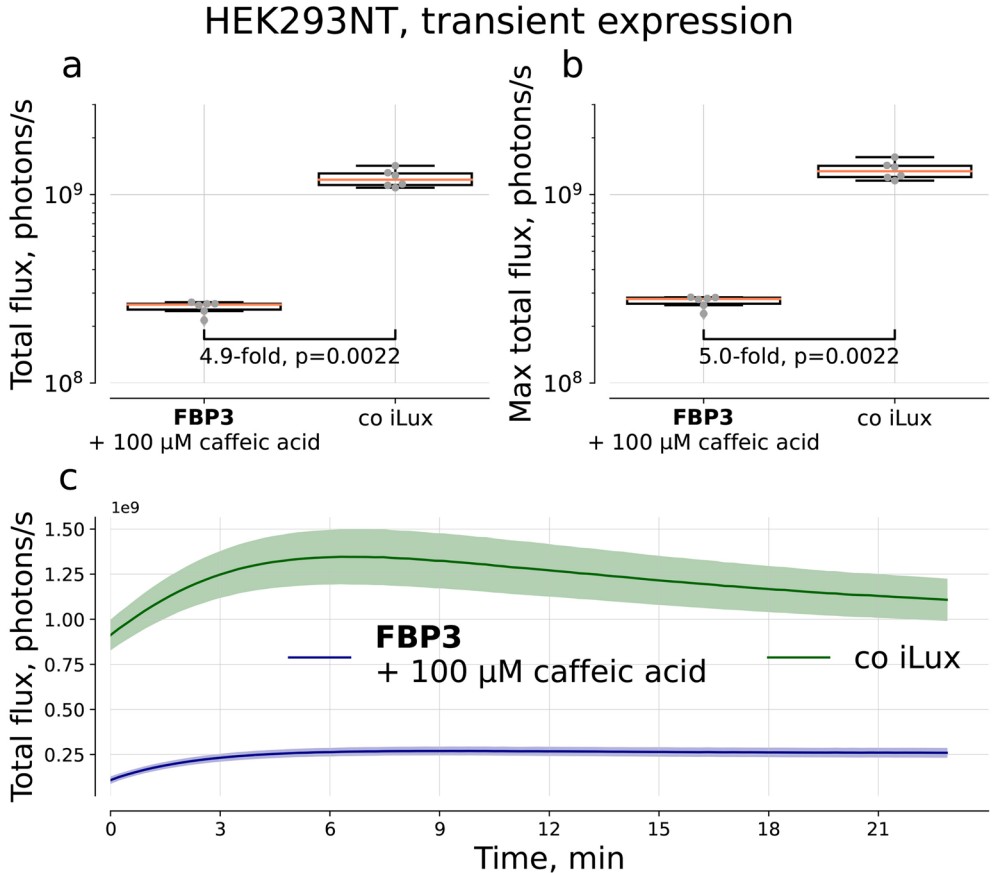

**Extended Data Fig. 10 | Comparison of autonomous bacterial bioluminescent system with FBP3 pathway after treatment with 100 µM caffeic acid in transient expression in HEK293NT.** (**a**) Average signal collected over 20 min. N = 6 biologically independent samples per box plot. (**b**) Maximum signal. The boxes are the first and the third quartiles, whiskers are the rest of the distribution except outliers, the orange line is the median. The difference between mean values and p-values of post-hoc two-sided Mann-Whitney U-tests are indicated below the brackets between the box plots (**c**) Kinetics. Data shown as mean (solid line) ± SD (area around the solid line).

# Reporting Summary

## Statistics

For all statistical analyses, confirm that the following items are present in the figure legend, table legend, main text, or Methods section.

| n/a | Confirmed | |
|---|---|---|
| ☐ | ☒ | The exact sample size (*n*) for each experimental group/condition, given as a discrete number and unit of measurement |
| ☐ | ☒ | A statement on whether measurements were taken from distinct samples or whether the same sample was measured repeatedly |
| ☐ | ☒ | The statistical test(s) used AND whether they are one- or two-sided<br>*Only common tests should be described solely by name; describe more complex techniques in the Methods section.* |
| ☒ | ☐ | A description of all covariates tested |
| ☐ | ☒ | A description of any assumptions or corrections, such as tests of normality and adjustment for multiple comparisons |
| ☐ | ☒ | A full description of the statistical parameters including central tendency (e.g. means) or other basic estimates (e.g. regression coefficient) AND variation (e.g. standard deviation) or associated estimates of uncertainty (e.g. confidence intervals) |
| ☐ | ☒ | For null hypothesis testing, the test statistic (e.g. *F*, *t*, *r*) with confidence intervals, effect sizes, degrees of freedom and *P* value noted<br>*Give P values as exact values whenever suitable.* |
| ☒ | ☐ | For Bayesian analysis, information on the choice of priors and Markov chain Monte Carlo settings |
| ☒ | ☐ | For hierarchical and complex designs, identification of the appropriate level for tests and full reporting of outcomes |
| ☒ | ☐ | Estimates of effect sizes (e.g. Cohen's *d*, Pearson's *r*), indicating how they were calculated |

*Our web collection on statistics for biologists contains articles on many of the points above.*

## Software and code

Policy information about availability of computer code

| Data collection | Python code for automated image acquisition. |
|---|---|
| Data analysis | FiJi ImageJ (ver. 1.53t), ImageJ plugin RankFilters (https://imagej.nih.gov/ij/source/ij/plugin/filter/RankFilters.java , ver. 2020-07-17), Living Image (ver. 4.5.5), Tecan SPARKCONTROL Dashboard (ver. 3.1 SP1), FusionCapt Advance Pulse 7 (ver. 17.03), LibRaw (ver. 0.19.2), custom Python code for image and data processing (Python ver. 3.10.12) (pandas (https://pandas.pydata.org/ , ver. 1.5.1) package, Seaborn (https://seaborn.pydata.org/ , ver. 0.12.2) package, matplotlib (https://matplotlib.org/ , ver. 3.7.1) package, numpy (https://numpy.org/ , ver. 1.23.5) package, scipy package (https://www.scipy.org/ , ver. 1.11.3). Scikit-posthocs Python package (https://pypi.org/project/scikit-posthocs/, ver. 0.8.0) for statistical analysis). |

For manuscripts utilizing custom algorithms or software that are central to the research but not yet described in published literature, software must be made available to editors and reviewers. We strongly encourage code deposition in a community repository (e.g. GitHub). See the Nature Portfolio guidelines for submitting code & software for further information.

## Data

Policy information about [availability of data](availability of data)

All manuscripts must include a [data availability statement](data availability statement). This statement should provide the following information, where applicable:

- Accession codes, unique identifiers, or web links for publicly available datasets
- A description of any restrictions on data availability
- For clinical datasets or third party data, please ensure that the statement adheres to our [policy](policy)

Data is available at https://doi.org/10.6084/m9.figshare.24623817. Python code for processing and plotting data is available at both https://github.com/Perfus/BL2.0 and https://doi.org/10.6084/m9.figshare.24623976.

## Human research participants

Policy information about [studies involving human research participants and Sex and Gender in Research.](studies involving human research participants and Sex and Gender in Research.)

| Reporting on sex and gender | n/a |
|---|---|
| Population characteristics | n/a |
| Recruitment | n/a |
| Ethics oversight | n/a |

Note that full information on the approval of the study protocol must also be provided in the manuscript.

# Field-specific reporting

Please select the one below that is the best fit for your research. If you are not sure, read the appropriate sections before making your selection.

☒ Life sciences    ☐ Behavioural & social sciences    ☐ Ecological, evolutionary & environmental sciences

For a reference copy of the document with all sections, see [nature.com/documents/nr-reporting-summary-flat.pdf](nature.com/documents/nr-reporting-summary-flat.pdf)

# Life sciences study design

All studies must disclose on these points even when the disclosure is negative.

| Sample size | The experiments described in this study were done for the first time. Due to exploratory nature of our study we refrained from unnecessary generalizations. No pre-specified effect size could be determined a priori. |
|---|---|
| Data exclusions | No data were excluded from the study. |
| Replication | Number of replicates are explicitly stated in the figure legends. Where applicable, reported results were consistently replicated across multiple experiments with all replicates generating similar results. |
| Randomization | No randomisation was performed. Our experiments had measurable and observable outcomes (e.g., enzyme activity, light emission etc). These outcomes are not influenced by subjective bias, making randomisation less relevant. |
| Blinding | No blinding was performed. Our experiments had measurable and observable outcomes (e.g., enzyme activity, light emission etc). These outcomes are not influenced by subjective bias, making blinding less relevant. |

# Reporting for specific materials, systems and methods

We require information from authors about some types of materials, experimental systems and methods used in many studies. Here, indicate whether each material, system or method listed is relevant to your study. If you are not sure if a list item applies to your research, read the appropriate section before selecting a response.

## Materials & experimental systems

| n/a | Involved in the study |
|---|---|
| ☒ | ☐ Antibodies |
| ☐ | ☒ Eukaryotic cell lines |
| ☒ | ☐ Palaeontology and archaeology |
| ☒ | ☐ Animals and other organisms |
| ☒ | ☐ Clinical data |
| ☒ | ☐ Dual use research of concern |

## Methods

| n/a | Involved in the study |
|---|---|
| ☒ | ☐ ChIP-seq |
| ☒ | ☐ Flow cytometry |
| ☒ | ☐ MRI-based neuroimaging |

## Eukaryotic cell lines

Policy information about cell lines and Sex and Gender in Research

| | |
|---|---|
| Cell line source(s) | HEK293 (ATCC), HEK293NT (unknown origin) |
| Authentication | Cell line was authenticated morphologically. |
| Mycoplasma contamination | Cell lines were frequently tested for mycoplasma contamination. Cell line used in this study was verified to be mycoplasma negative before undertaking experiments with it. |
| Commonly misidentified lines (See ICLAC register) | No commonly misidentified cells were used. All cells displayed homogeneous characteristic morphology. |

