## [Peer Review File · Nature Methods]

Peer Review Information

Manuscript Title: An improved pathway for autonomous bioluminescence imaging in eukaryotes

Corresponding author name(s): Karen S. Sarkisyan, Alexander S. Mishin

Editorial Notes: None

Reviewer Comments & Decisions:

Decision Letter, initial version:

Dear Karen,

Your Brief Communication, "An improved pathway for autonomous bioluminescence imaging in eukaryotes", has now been seen by three reviewers. As you will see from their comments below, although the reviewers find your work of considerable potential interest, they have raised a number of concerns. We are interested in the possibility of publishing your paper in Nature Methods, but would like to consider your response to these concerns before we reach a final decision on publication. We therefore invite you to revise your manuscript to address these concerns.

In your revision, we would like it to be more clear what each of the changes to the system is contributing to the overall performance boost. In terms of in vitro characterizations, we do think it would be appropriate for you to characterize the intrinsic brightness of the luciferin/luciferase, and kinetic properties of the enzymes (if they can be purified--if not, please let me know).

Please clarify all questions from the reviewers regarding expression levels and constructs, so it's clear that all comparisons were done as fairly as possible. We would like to see a comparison to the bacterial system mentioned by referee 1, but do not require any in vivo mouse work as lightly suggested by reviewer 2 (though we think the potential for applications in this space should be discussed). When you revise, we ask that you don't overstate the performance across species but note that the clearest benefits are currently for plant expression.

This URL links to your confidential home page and associated information about manuscripts you may have submitted, or that you are reviewing for us. If you wish to forward this email to co-authors, please delete the link to your homepage.

We hope to receive your revised paper within three months. If you cannot send it within this time, please let us know. In this event, we will still be happy to reconsider your paper at a later date so long as nothing similar has been accepted for publication at Nature Methods or published elsewhere.

OPEN SCIENCE REQUIREMENTS

REPORTING SUMMARY AND EDITORIAL POLICY CHECKLISTS

IMAGE INTEGRITY

DATA AVAILABILITY

All novel DNA and RNA sequencing data, protein sequences, genetic polymorphisms, linked genotype and phenotype data, gene expression data, macromolecular structures, and proteomics data must be

deposited in a publicly accessible database, and accession codes and associated hyperlinks must be provided in the “Data Availability” section.

Please include a “Data availability” subsection in the Online Methods. This section should inform readers about the availability of the data used to support the conclusions of your study, including accession codes to public repositories, references to source data that may be published alongside the paper, unique identifiers such as URLs to data repository entries, or data set DOIs, and any other statement about data availability. At a minimum, you should include the following statement: “The data that support the findings of this study are available from the corresponding author upon request”, describing which data is available upon request and mentioning any restrictions on availability. If DOIs are provided, please include these in the Reference list (authors, title, publisher (repository name), identifier, year). For more guidance on how to write this section please see: <http://www.nature.com/authors/policies/data/data-availability-statements-data-citations.pdf>

MATERIALS AVAILABILITY

SUPPLEMENTARY PROTOCOL

To help facilitate reproducibility and uptake of your method, we ask you to prepare a step-by-step Supplementary Protocol for the method described in this paper. We [encourage authors to share their step-by-step experimental protocols](https://www.nature.com/nature-research/editorial-policies/reporting-standards#protocols) on a protocol sharing platform of their choice and report the protocol DOI in the reference list. Nature Portfolio 's Protocol Exchange is a free-to-use and open resource for protocols; protocols deposited in Protocol Exchange are citable and can be linked from the published article. More details can found at www.nature.com/protocolexchange/about.

ORCID

Sincerely,
Rita

Rita Strack, Ph.D.
Senior Editor
Nature Methods

Reviewers' Comments:

Reviewer #1:

Remarks to the Author:

Shakhova et al. describe the optimization of the fungal bioluminescent pathway in different eukaryotic expression systems (plant, yeast and mammalian). By combining mutagenesis of nnLuz and nnH3H, screening of HispS from different fungi and coexpression of NpgA, bioluminescence emission is improved by 1-2 orders of magnitude. A particular focus of the work is on the generation of different plant species with enhanced luminescence which largely seems to originate from the incorporation of NpgA (e.g. Figure 1b and Extended Data Figure 2), as also described in a very recent publication (doi: 10.1111/pbi.14068). In mammalian cells, luminescence emission still depends on the addition of caffeic acid as a luciferin precursor so that fully autonomous bioluminescence emission is not yet achieved. Nevertheless, the described improvements of several enzymes of the fungal bioluminescent system represent a significant progress towards this goal and will therefore be of great interest to a broad readership.

The manuscript is clearly written and contains comprehensive and technically sound data that describe the obtained improvements in detail using appropriate methodology and statistics. The following points should be addressed before publication:

1. The influence of nnLuz_v4 and nnH3H_v2, the incorporation of NpgA and the replacement of nnHispS by mcitHispS on the overall light emission in the different expression systems is not fully clear. To allow a better evaluation, in each of Figures 1b-d the following variants should be compared (indicating the fold improvements in luminescence relative to FBP1 in each case): FBP1, FBP1+nnH3H_v2/nnLuz_v4, FBP2 and FBP3. Numbers indicating the fold change in luminescence between FBP1 and FBP1+nnH3H_v2/nnLuz_v4 should also be included in other figures (Figure 2a, Extended Data Figure 5, Supplementary Figures 9a,c and 12b).
2. A major advantage of the fungal bioluminescent system is that it potentially enables autonomous bioluminescence emission in different eukaryotic hosts also other than plants, most importantly mammalian cells. Although addition of caffeic acid still seems to be a requirement here, a direct comparison of the luminescence from FBP3 (+caffeic acid) to the bacterial bioluminescent system which already allows fully autonomous bioluminescence emission would be of interest (described for instance in FEMS Yeast Res 4, 305-313 (2003) for yeast and in PNAS 116, 26491-26496 (2019) for mammalian cells). In this way, the autonomous luminescence intensity that could ultimately be achieved with the developed FBP3 enzymes (once cellular caffeic acid synthesis is established) can be better evaluated.
3. In Extended Data Figures 8 and 9, the luciferins and their concentrations should be given for each luciferase.

4. In the plasmids for the creation of transgenic plant lines, the genes nnLuz, nnCPH and nnH3H have been rearranged between FBP1 and FBP1+nnLuz_v4/nnH3H_v2 (Extended Data Figure 4). Does this rearrangement affect the luminescence emission?
5. Why are different designations for the luminescence signal used in the figures throughout the manuscript (e.g. Figure 1b-d)? Where possible, axes should be labeled consistently.
6. Page 1, line 23: Bacterial bioluminescence is not restricted to marine bacteria, but is also found in some terrestrial and freshwater bacteria.
7. Figure 1: The concentration of caffeic acid should be stated in the legend (where added).
8. Supplementary Figures 1 and 2: At what temperature were the measurements performed?
9. Supplementary Figure 7: Why is the ratio of hispidin RLU / luciferin RLU shown (instead of hispidin RLU only)? Coexpression of the nnLuz variant should be stated in the legend.
10. Supplementary Figure 8: y-axis should be labeled with at least two numbers to determine the scale.

Reviewer #2:

Remarks to the Author:

Shakhova et al. describe new bioluminescence imaging tools by engineering enzymes in fungal bioluminescence pathway. They focused on enzymes that catalyzes light-emitting caffeic acid cycle in the fungus *Neonothopanus nambi*, which has recently attracted attention for developing reporter tools for transient expression assays in plants. By introducing substitution mutations in the enzymes, they successfully increased enzyme stability and achieved brighter luminescence by 2 sets of mutated enzyme combination: FBP2 and FBP3. Surprisingly, the authors also showed that the luminescence of plants can be observed using a consumer camera and smartphone cameras. These sets of enzymes can broaden the applicability of self-sustained luminescence for plant biology. Furthermore, caffeic acid was able to produce luminescence using FBP3 in HEK293 cells although its signal intensity was much less than those produced by luciferin-firefly luciferase and luciferin-NanoLuc reactions.

This study on the use of fungal bioluminescence pathway is of potential interest and seems to be useful for developing bioluminescence reporter tools. However, the reviewer felt a lack of information on superiority and versatility compared to existing imaging systems. The reviewer is particularly concerned regarding the following points

The newly constructed high-intensity reporter system can be highly evaluated for realizing simple and highly sensitive in-vivo imaging in plants. On the other hand, the usefulness of the constructed new system has not been sufficiently demonstrated:

1) The authors focused on the light-emitting caffeic acid cycle in the fungus *Neonothopanus nambi*, an improved version of the recently described luminescence system. not their original. In addition, methods for improvement include the introduction of substitutional mutations using existing information and the introduction of random mutations, and methods newly developed by them are not used.

2) Highly sensitive optical in vivo imaging has been achieved by fluorescence imaging. The multiplicity of fluorescence makes it possible to visualize many molecules simultaneously, which is useful for elucidating their roles and interactions. From a methodology point of view, they need to demonstrate the superiority of this system by showing results that can only be observed using their luminescence system.

3) The high luminescence intensity has unfortunately not been obtained except for plant. Although they certainly observed luminescence in human cells, any images were provided. Fluorescence provides images with better resolution than luminescence. Therefore, this reviewer cannot come up with a methodology in which luminescence outperforms fluorescence in observations at the level of cultured cells. If this system can be applied to small animals, it will be very valuable as a new optical imaging system.

Minor comment;

1) The authors use uM instead of μM throughout the manuscript. It should be revised.

2) Fig. S9d does not seem to reflect the result of Fig. S9c (all combinations are $p < 0.001$). Is Fig. S9d correct?

Reviewer #3:

Remarks to the Author:

Bioluminescence had been used for imaging, and the authors have reported modifications to the fungal bioluminescent system (FBP2 and FBP3) that increased the autoluminescence intensity by up to 1-2 orders of magnitude compared to the wild-type. The study extends from the authors previous report using the fungal autoluminescence system in multiple hosts (Ref), and enhancing the luminescence intensity is an important aspect to use this system as a tool for biological studies. The data seems to point to the inclusion of NpgA as the most substantial contributor to enhanced luminescence, and I felt that this should be discussed more, including the mechanisms of how the mutation of Luz and H3H

contributed to enhanced luminescence. In addition, I have several concerns that would be better explained for the readers to understand the study.

(1) The authors describe that the newly developed bioluminescence (FBP2 and FBP3) has much higher luminescence intensity than the wild-type (FBP1), however, the data does not always include all three systems depending on the experiment host, and this makes it difficult to understand the relative luminescence intensity against FBP1.

(2) When luminescence intensity is compared for the Luz mutants using *E. coli*, how was the luminescence normalized? I suspect that the protein expression level and possibly the characteristics of the protein (e.g. solubility, codon preference) may be affected by the mutation, which will influence the luminescence intensity.

(3) In Fig. 1b, the fold-change of FBP2 vs. FBP3 is not stated. From a glance, this fold-change seems to be much greater than the difference of nnHisP vs. mcitHisP in Ext. data 1a, despite both data compared nnHisP and mcitHisP in BY-2 cells. Is there a reason for this difference in luminescence fold-change?

(4) The authors compare FBP luminescence with Nanoluc and Fluc (Extended data Fig. 8 and 9).

However, the conversion efficiency of caffeic acid to luciferin and the actual cellular concentration of the substrates are not described. In addition, each system uses a substrate with different chemical structure/properties, and I am unsure whether the cellular permeability is the same (caffeic acid for FBP). Similarly, the kinetics depends on the concentration of the luciferase and luciferin. Taken together, I feel it is difficult to compare the luminescent intensity by this method, and would like to suggest *in vitro* comparison using purified proteins, or a similar approach that can be normalized to the quantity of the luciferase.

(5) The fold-increase achieved in this study is remarkable, however, there are no insights into the mechanism of the enhanced luminescence. In relation to the mutagenesis of Luz, H3H, or the activity of mcitHisP, or NpgA it would be interesting to see whether the enzymatic properties (e.g. kcat, quantum yield) caused the enhancement.

(6) From the transient assay of NpgA in plant cells (Extended Fig. 2), the inclusion of NpgA seemed to have the greatest effect (up to 280-fold increase in petunia) on the luminescence intensity among all of the modifications presented in this study. In stable transgenic plants, 8- and 54-fold increase was reported for *N. benthamiana* and *N. tabacum*, respectively. I feel that this was the largest contributor that enhanced luminescence and is worth discussing about the possible reasons. For example, was there a difference in transgene expression level? FBPs protein activity? Endogenous caffeic acid level?

(7) When comparing the different FBP systems (FBP1, FBP2, FBP3), can the transgene expression level of each gene influence the luminescence intensity? Can the transgene expression level change depend on the order of the transgenes within the multiple expression cassette (e.g. possible transgene silencing triggered depending on the position of the transgene in the cassette)? Perhaps, a RT-qPCR to check whether there are any differences in transgene expression level of the different constructs may help. Or confirming the expression of each protein may also help.

(8) FBP3 was not brighter than FBP2 in stable transgenic *N. benthamiana* lines, which is different to the BY-2, mammalian, yeast cells. Is there any reason for this?

(9) When comparing FBP1 to FBP2 or FBP3 in luminescent plants (Figure 2, Extended data Fig.6, Supplementary Fig. 12, Supplementary Fig. 14-15, Supplementary Fig. 17-18), which generation of the plant was used? And if different generations were compared, was the luminescence intensity stable over generations?

(10) nnLuz v4 did not show any enhancement of luminescence intensity in yeast (Supplementary Fig. 8). Is there any reason for this, and I feel it would help the readers if this was discussed.

(11) For the HEK293 and pichia assay comparing FBP1 and FBP3 (Supplementary Fig. 10 ad 11), was the FBP1 co-transfected with NpgA and FBP3 in a single vector? If this is the case, could the co-transfection efficiency affect the luminescence intensity in these data? and the data would be inconclusive about how much NpgA or H3H_v2 contributed to the luminescence intensity?

(12) The effect of NpgA was negligible in *N. benthamiana* transgenic lines (Supplementary Fig.14), but large in *N. tabacum*. Is there any reason for this?

Minor points

Page 1: In the sentence, “ Low enzymatic activity and limited stability of enzymes at physiologically relevant temperatures 2 resulted in modest light output...” Should this be quantitative? It may be better to briefly describe the limitations of the FBP1 in relation to its application (e.g. imaging) for the readers to understand the benefit of the new system easier.

Page 2: In the sentence “Similarly to nnHisS, mcitHisS was efficiently activated by phosphopantetheinyl transferase NpgA from *Aspergillus nidulans*, which we confirmed to be a necessary component for bioluminescence in most plant species”, the FBP system has been tested mainly on dicot species and it may be an overstatement to say “most plant species”. Otherwise, if it can be explained that most plant species do not have phosphopantetheinyl transferase, it may clarify the point.

Page 2: For the sentence “When stably expressed from a genomic copy, the wild-type fungal pathway FBP1 performed well in tobacco species, however, in our hands it did not yield sufficiently bright luminescence in other species.”, depending on the imaging system, and its purpose, dim luminescence can still be useful. I was not able to follow what would be the required brightness for what application.

Page 2: For the sentence “Furthermore, the brightest tissues – petunia flower buds – could be recorded on modern smartphone cameras.”, I could not find the image data taken using smartphone cameras.

Tested on *benthamiana*, petunia BY-2. Maybe overstatement to say most plant species, especially without demonstrating on monocots.

Figure 1 caption: The comparison of FBP2 is lacking in mammalian and yeast cells. Was there any reason for this, and if FBP2 is to be omitted, the sentence should be rephrased to clarify this.

Figure 1e: It would help the readers to see the difference of luminescence intensity between FBP1, FBP2 and FBP3 as an image for comparison of brightness. This would be important as the comparison in the graph has used tobacco BY-2 cells, which may be different to petunia.

Figure 2: Because the ISO range that was used is large, it would help the readers to understand the relative brightness of each plant by adding the details about the imaging conditions for each species (e.g. ISO, distance to the subject (if they were different for each species)).

Extended Fig. 3: Please add the fold-difference compared to the wild-type for understanding the contribution of the different HispS and H3H to the luminescence intensity.

Extended Fig. 7: It would also be nice to show the wild-type (non-transgenic) plants for phenotypic analysis.

Supplementary Fig. 4b (right-side of the graph): How was the “Expression” determined? Gene expression or protein expression level?

Supplementary Fig. 5: Was there any reason the Supplementary Fig. 3 used *E. coli* and Supplementary Fig 5. used *Pichia*? Also, if there are no particular reasons, consistent presentation style may be easier for viewing (i.e. line graph or box and whisker plot).

Supplementary Fig. 17 and 18: The two figures are somewhat redundant in the sense that the only difference is the age of the plant. Perhaps it may be better to combine the graphs so it is more obvious that the luminescence intensity has gained over development.

Graphs in general: I felt that the font size relative to the graph was small, and sometimes difficult to read.

Author Rebuttal to Initial comments

Reviewer #1:

Remarks to the Author:

Shakhova et al. describe the optimization of the fungal bioluminescent pathway in different eukaryotic expression systems (plant, yeast and mammalian). By combining mutagenesis of nnLuz and nnH3H, screening of HispS from different fungi and coexpression of NpgA, bioluminescence emission is improved by 1-2 orders of magnitude. A particular focus of the work is on the generation of different plant species with enhanced luminescence which largely seems to originate from the incorporation of NpgA (e.g. Figure 1b and Extended Data Figure 2), as also described in a very recent publication (doi: 10.1111/pbi.14068). In mammalian cells, luminescence emission still depends on the addition of caffeic acid as a luciferin precursor so that fully autonomous bioluminescence emission is not yet achieved. Nevertheless, the described improvements of several enzymes of the fungal bioluminescent system represent a significant progress towards this goal and will therefore be of great interest to a broad readership.

The manuscript is clearly written and contains comprehensive and technically sound data that describe the obtained improvements in detail using appropriate methodology and statistics. The following points should be addressed before publication:

1. The influence of nnLuz_v4 and nnH3H_v2, the incorporation of NpgA and the replacement of nnHispS by mcitHispS on the overall light emission in the different expression systems is not fully clear. To allow a better evaluation, in each of Figures 1b-d the following variants should be compared (indicating the fold improvements in luminescence relative to FBP1 in each case): FBP1, FBP1+nnH3H_v2/nnLuz_v4, FBP2 and FBP3.

We agree with the reviewer that a demonstration of influence of individual enhancements on the overall luminescence intensity would improve the manuscript. We performed additional experiments in yeast and mammalian cells (updated Figure 1) to address the issue.

Extended Data Figure 2 shows the effect of co-expression of NpgA in plant systems. Extended Data Figure 3 contains information on performance of all combinations of nnLuz, nnH3H and HispS variants in plant cells. Similarly, we now report an extended comparison of variants of the bioluminescence pathway in mammalian cells (updated Figure 1c) and in yeast (updated Figure 1d).

We now discuss these results and the relative contribution of improvements in the revised version of the manuscript. Overall, the major improvement in brightness is due to co-expression with NpgA, since it is what determines the functionality of hispidin synthase in the absence of endogenous PPTase activity. Relative contributions of other enzymes depend on the host. In yeast, improved hispidin hydroxylase seems to have the largest effect. In mammalian cells it is the luciferase. In plant cells, hispidin synthase and hispidin hydroxylase have comparable contributions.

Possible reasons for differences between heterologous systems might include:

- the level of endogenous PPTase activity, which determines relative contribution of NpgA,
- growth conditions, such as temperature,
- copy numbers of bioluminescence genes,
- cell biology of the host organism: endogenous competing enzymes, differences in subcellular distribution of bioluminescence enzymes, presence of inhibitors, etc.

Numbers indicating the fold change in luminescence between FBP1 and FBP1+nnH3H_v2/nnLuz_v4 should also be included in other figures (Figure 2a, Extended Data Figure 5, Supplementary Figures 9a,c and 12b).

We now report fold change values on figures where appropriate.

2. A major advantage of the fungal bioluminescent system is that it potentially enables autonomous bioluminescence emission in different eukaryotic hosts also other than plants, most importantly mammalian cells. Although addition of caffeic acid still seems to be a requirement here, a direct comparison of the luminescence from FBP3 (+caffeic acid) to the bacterial bioluminescent system which already allows fully autonomous bioluminescence emission would be of interest (described for instance in FEMS Yeast Res 4, 305-313 (2003) for yeast and in PNAS 116, 26491-26496 (2019) for mammalian cells). In this way, the autonomous luminescence intensity that could ultimately be achieved with the developed FBP3 enzymes (once cellular caffeic acid synthesis is established) can be better evaluated.

We agree with the reviewer's suggestion. In fact, we think that the bacterial system was overlooked as a useful reporter by the community, possibly due to early claims about its toxicity to eukaryotic cells.

We have performed the comparison of the bacterial and fungal systems in mammalian and plant cells.

For mammalian expression, we ordered the optimised nucleotide sequences of bioluminescence genes exactly as reported by Gregor et al. and cloned these into our vectors. Transfection of HEK293NT cells with these plasmids along with FBP3 (+caffeic acid) allowed detection of bioluminescence. The brightness of bioluminescence of the bacterial bioluminescence was 5-fold brighter than the FBP3 (Extended Figure 10).

In mammalian systems, the fungal bioluminescence pathway still requires optimisation – which will be the focus of our future work. However, in contrast to the wild type fungal bioluminescence system, both bacterial and fungal pathways now confer easily detectable bioluminescence in animal cells – and their orthogonality could potentially be used for multi-colour autoluminescence imaging. We now discuss suboptimal performance of the fungal pathway in mammalian cells in the manuscript.

For plant expression (Extended Figure 9), we assembled two versions of the bacterial bioluminescence pathway: in one, genes were expressed in the cytoplasm, and in another one genes were targeted to plastids due to a potentially larger pool of fatty acid precursors in this compartment. We assessed two versions of the bacterial pathway for each localisation: all essential genes (*luxA*, *luxB*, *luxC*, *luxD*, *luxE*) with and without co-expression with the *frp* gene encoding flavin reductase.

In BY-2 cells, depending on gene dosage tested and localisation, the fungal bioluminescent system emitted 2-4 orders of magnitude more light than the bacterial pathway. In plant leaves, we only tested a single gene dosage, with the fungal system outperforming the bacterial one by over two orders of magnitude.

3. In Extended Data Figures 8 and 9, the luciferins and their concentrations should be given for each luciferase.

We added the information about substrate concentration (the current numbering is Extended Data Figures 7 and 8). For furimazine, as the concentration is not stated in the kit documentation, we indicated the volume of formazin solution added.

4. In the plasmids for the creation of transgenic plant lines, the genes nnLuz, nnOPH and nnH3H have been rearranged between FBP1 and FBP1+nnLuz_v4/nnH3H_v2 (Extended Data Figure 4). Does this rearrangement affect the luminescence emission?

The plasmids used to create transgenic lines indeed had different structures. We did this for technical reasons under the initial assumption that the order of transcription units would not significantly affect luminescence. However, a recent paper from Patron and Orzaez groups showed that construct architecture influences expression and product yield in plants.

To allow for fair comparison, we assembled new plasmids encoding FBP1 genes with exactly the same regulatory elements and the same order of transcription units as in FBP2 and FBP3 plasmids. When such harmonised plasmids were compared in BY-2 cells, we observed a stepwise increase in luminescence as more improved genes were introduced into the plasmid (updated Figure 1d). These results confirm that observed changes in luminescence are due to gene improvements rather than altered order of transcription units.

Similar conclusions could be drawn from experiments where plasmids – each encoding a single transcription unit – were co-transfected (such as experiments in BY-2 on Extended Data Fig. 3 and mammalian cells and yeast reported on Figure 1). These experiments confirm that major improvements in the pathway performance do not come from different expression levels of individual transcription units.

However, the order of genes does affect luminescence. For this revision, we also assembled new FBP2-like and FBP3-like plasmids, changing the order of genes to correspond to the original FBP1 plasmid. We observed 1.5 and 1.8-fold drop in luminescence for FBP2 and FBP3, respectively (Supplementary Figure 10b).

We believe that these results are in accordance with data by Kallam et al., and provide an interesting starting point to investigate the relationship between gene dosage, construct architecture and product yield. At the same time, they do not significantly change the conclusions reported in our paper.

5. Why are different designations for the luminescence signal used in the figures throughout the manuscript (e.g. Figure 1b-d)? Where possible, axes should be labeled consistently.

We aimed to display absolute photon counts when it was possible to obtain this information on our equipment. We aimed to show that, as this information may become useful for comparison between experiments performed in different labs.

We harmonised inconsistent labelling of Y axes on Figure 1 and other figures, where appropriate.

6. Page 1, line 23: Bacterial bioluminescence is not restricted to marine bacteria, but is also found in some terrestrial and freshwater bacteria.

Thank you, corrected.

7. Figure 1: The concentration of caffeic acid should be stated in the legend (where added).

We now state the concentration of caffeic acid in the legend.

8. Supplementary Figures 1 and 2: At what temperature were the measurements performed?

Experiments in plants and yeast in the **Supplementary Figures 1 and 2** were performed at room temperature, experiments in mammalian cells – at 37°C. We now state the temperature of experiments in the legend.

9. **Supplementary Figure 7: Why is the ratio of hispidin RLU / luciferin RLU shown (instead of hispidin RLU only)?**
Coexpression of the nnLuz variant should be stated in the legend.

The ratio of hispidin / luciferin RLUs was used as a readout to account for possible noise in luciferase expression. We now state the nnLuz variant in the legend (the new numbering is **Supplementary Figure 8**).

10. **Supplementary Figure 8: y-axis should be labeled with at least two numbers to determine the scale.**

Thank you for pointing this out, now corrected.

Reviewer #2:

Remarks to the Author:

Shakhowa et al. describe new bioluminescence imaging tools by engineering enzymes in fungal bioluminescence pathway. They focused on enzymes that catalyzes light-emitting caffeic acid cycle in the fungus *Neonothopanus nambi*, which has recently attracted attention for developing reporter tools for transient expression assays in plants. By introducing substitution mutations in the enzymes, they successfully increased enzyme stability and achieved brighter luminescence by 2 sets of mutated enzyme combination: FBP2 and FBP3. Surprisingly, the authors also showed that the luminescence of plants can be observed using a consumer camera and smartphone cameras. These sets of enzymes can broaden the applicability of self-sustained luminescence for plant biology. Furthermore, caffeic acid was able to produce luminescence using FBP3 in HEK293 cells although its signal intensity was much less than those produced by luciferin-firefly luciferase and luciferin-NanoLuc reactions.

This study on the use of fungal bioluminescence pathway is of potential interest and seems to be useful for developing bioluminescence reporter tools. However, the reviewer felt a lack of information on superiority and versatility compared to existing imaging systems. The reviewer is particularly concerned regarding the following points

The newly constructed high-intensity reporter system can be highly evaluated for realizing simple and highly sensitive in-vivo imaging in plants. On the other hand, the usefulness of the constructed new system has not been sufficiently demonstrated:

1) The authors focused on the light-emitting caffeic acid cycle in the fungus *Neonothopanus nambi*, an improved version of the recently described luminescence system, not their original. In addition, methods for improvement include the introduction of substitutional mutations using existing information and the introduction of random mutations, and methods newly developed by them are not used.

We apologise for possible confusion. Indeed, we used consensus mutagenesis, random mutagenesis, and comparison of homologues, to identify a version of the fungal bioluminescence pathway with improved performance.

2) Highly sensitive optical in vivo imaging has been achieved by fluorescence imaging. The multiplicity of fluorescence makes it possible to visualize many molecules simultaneously, which is useful for elucidating their roles and interactions. From a methodology point of view, they need to demonstrate the superiority of this system by showing results that can only be observed using their luminescence system.

We agree with the reviewer that outstanding progress has been achieved with a variety of fluorescence live imaging approaches over the past decades. Similarly, in the field of luminescence, high-performing engineered luciferases have been developed to monitor physiology when exogenous substrate is provided.

The downsides of fluorescence-based approaches include (1) requirement for illumination with excitation light, (2) autofluorescence of tissues, (3) photobleaching. In heavily pigmented plant tissues, this autofluorescence is especially limiting, becoming "a nightmare for many fluorescence applications" (Handbook of biological confocal microscopy, 2006). Luminescence is more tolerant to presence of pigments and allows for a much wider dynamic range due to absence of the background signal, but the requirement for addition of the luciferin makes the imaging invasive, potentially toxic, and expensive. In addition, luciferins often suffer from poor tissue permeability, poor transport between the tissues, and degradation.

Pathways for *autoluminescence* enable non-invasive physiology imaging without the need for exogenously applied substrate. This enables longitudinal non-invasive imaging studies of potentially any physiological event. It also potentially allows experiments outside of the laboratory setting (for example, in the ecologically relevant context of a meadow or a forest).

In our previous work, we showed proof-of-concept application of the fungal bioluminescence pathway in eukaryotes, however, it turned out that the performance of the pathway outside of the tobacco species tested in that work was poor. That made it difficult to use FBP as a molecular tool – and in the current work we aimed to improve the pathway.

We believe that improved versions of the pathway reported in this manuscript enable broad application of autoluminescence across plant and likely fungal species. As shown in the response to comments of Reviewer 1, the performance of the pathway in animal systems remains suboptimal and will be the focus of our future work.

3) The high luminescence intensity has unfortunately not been obtained except for plant. Although they certainly observed luminescence in human cells, any images were provided. Fluorescence provides images with better resolution than luminescence. Therefore, this reviewer cannot come up with a methodology in which luminescence outperforms fluorescence in observations at the level of cultured cells. If this system can be applied to small animals, it will be very valuable as a new optical imaging system.

We used cultured cells as a system to prototype and assess versions of the bioluminescence pathway. We believe that the major application of autoluminescence lies in physiology imaging of *multicellular* eukaryotes. We agree with the reviewer that it would be ideal to have the pathway working robustly in small animals.

Regarding the point about the intensity of luminescence, we now added a demonstration of high brightness of luminescent yeast in caffeic-acid-containing media – with luminescence intensities exceeding those of plants. We provide iPhone-shot photos and videos of the yeast culture as **Supplementary Figure 26** and **Supplementary Video 1**.

Although they certainly observed luminescence in human cells, any images were provided.

Unfortunately, we do not have access to a *luminescence* microscope needed to obtain high-quality single-cell images, and thus provide just the luminescence intensity values calculated from the images obtained on IVIS Spectrum, like the one below:

Therefore, this reviewer cannot come up with a methodology in which luminescence outperforms fluorescence in observations at the level of cultured cells. If this system can be applied to small animals, it will be very valuable as a new optical imaging system.

We agree with the reviewer that it would be very valuable to use autoluminescence in animals. At this point, we believe that the fungal pathway requires further optimisation before its performance justifies the resources and ethical considerations needed to create and maintain transgenic animals.

Regarding cultured cells and cell-based assays, an example of such methodology could be high-content screening in drug discovery, where luminescence enables quantitative measurements over a wide dynamic range which typically spans 5-6 orders of magnitude.

Minor comment;

1) The authors use uM instead of μM throughout the manuscript. It should be revised.

Thank you for pointing that out. We corrected the notation.

2) Fig. S9d does not seem to reflect the result of Fig. S9c (all combinations are $p < 0.001$). Is Fig. S9d correct?

We double checked, and the p-value is correct. The exact p-value is 7.9e-04 (Supplementary Figure 10b).

Reviewer #3:

Remarks to the Author:

Bioluminescence had been used for imaging, and the authors have reported modifications to the fungal bioluminescent system (FBP2 and FBP3) that increased the autoluminescence intensity by up to 1-2 orders of magnitude compared to the wild-type. The study extends from the authors previous report using the fungal autoluminescence system in multiple hosts (Ref), and enhancing the luminescence intensity is an important aspect to use this system as a tool for biological studies. The data seems to point to the inclusion of NpgA as the most substantial contributor to enhanced luminescence, and I felt that this should be discussed more, including the mechanisms of how the mutation of Luz and H3H contributed to enhanced luminescence. In addition, I have several concerns that would be better explained for the readers to understand the study.

We agree with the reviewer. We performed additional experiments (as outlined in response to the first point of Reviewer #1), and we now discuss in the manuscript the role of NpgA and the relative contributions of pathway components to overall performance boost.

(1) The authors describe that the newly developed bioluminescence (FBP2 and FBP3) has much higher luminescence intensity than the wild-type (FBP1), however, the data does not always include all three systems depending on the experiment host, and this makes it difficult to understand the relative luminescence intensity against FBP1.

We now performed additional experiments to address relative contributions of improvements to pathway components, which we provide on Figure 1, and discuss (along with data on Extended Data Figure 2 & 3) in the main text.

(2) When luminescence intensity is compared for the Luz mutants using E.coli, how was the luminescence normalized? I suspect that the protein expression level and possibly the characteristics of the protein (e.g. solubility, codon preference) may be affected by the mutation, which will influence the luminescence intensity.

We did not normalise protein expression levels in E.coli, and we agree with the reviewer that multiple reasons, including translation efficiency and solubility could have contributed to improvements observed during the directed evolution.

However, HiBit-based normalisation to protein abundance levels of selected mutants upon expression in mammalian cells allowed us to conclude that the observed improvements in luminescence were not reflecting codon preferences or other E.coli-specific effects.

(3) In Fig. 1b, the fold-change of FBP2 vs. FBP3 is not stated. From a glance, this fold-change seems to be much greater than the difference of nnHisP vs. mcitHisP in Ext. data 1a, despite both data compared nnHisP and mcitHisP in BY-2 cells. Is there a reason for this difference in luminescence fold-change?

We now state the fold-changes on the updated Figure 1. Regarding the difference in the fold change values between Figure 1 and Extended Data Figure 1, the difference is due to the use of *wild type* versions of bioluminescence enzymes in experiments reported on Extended Data Figure 1.

On the Extended Data Figure 1, wild type versions of other enzymes were used in the assay to compare hispidin synthases, while on Figure 1 we provided comparison of all-enzyme sets, including improved hispidin hydroxylase and luciferase. We now added new data to Figure 1 and provided information about the background enzymes to the legend of Extended Data Figure 1.

(4) The authors compare FBP luminescence with Nanoluc and Fluc (Extended data Fig. 8 and 9). However, the conversion efficiency of caffeic acid to luciferin and the actual cellular concentration of the substrates are not described. In addition, each system uses a substrate with different chemical structure/properties, and I am unsure whether the cellular permeability is the same (caffeic acid for FBP). Similarly, the kinetics depends on the concentration of the luciferase and luciferin. Taken together, I feel it is difficult to compare the luminescent intensity by this method, and would like to suggest *in vitro* comparison using purified proteins, or a similar approach that can be normalized to the quantity of the luciferase.

We agree that our experiments have not allowed comparison of luminescence intensity in excess of substrates and co-factors, and have not yielded data to quantitatively compare enzymatic properties of pathway components. In our experiments, comparison with Nanoluc and firefly luciferase was meant to serve as a rough benchmarking exercise, hopefully useful for scientists that used previously developed luciferases. The comparison was not meant to be a comprehensive comparison of catalytic activity.

We believe that assessment of *autoluminescence* pathways makes most sense in the context of host metabolism. In this work, by focusing on *in vivo* experiments, we hoped to provide practically useful data that could guide application of the pathway by other researchers. Following reviewers' recommendations, we now provide a direct comparison with another autonomous luminescence system – bacterial bioluminescence pathway – in two hosts: plant and mammalian cells (Extended Data Figures 9, 10).

(5) The fold-increase achieved in this study is remarkable, however, there are no insights into the mechanism of the enhanced luminescence. In relation to the mutagenesis of Luz, H3H, or the activity of *mcitHisps*, or *NpgA* it would be interesting to see whether the enzymatic properties (e.g. *k_{cat}*, quantum yield) caused the enhancement.

We do share the interest in mechanistic understanding of individual effects of mutations towards expression, folding and catalytic properties of the enzymes. We attempted to do such experiments, but ran into significant technical hurdles trying to purify all the versions of enzymes in the active state. For hispidin synthase, activity was lost significantly upon purification. For the luciferase, the purification of the active enzyme was difficult due to its transmembrane domain. We are continuing the work towards in-depth biochemical characterisation of bioluminescence enzymes *in vitro* but we believe it would be best completed outside of the timeline of this revision.

For this revision, we managed to perform lysate-based quantification of bioluminescence activity. We created yeast strains expressing HiBit-tagged luciferase to allow for normalisation of protein abundance, and analysed luminescence at different concentrations of luciferin. The results show that *nnLuz_v4* shows higher *V_{max}* and slightly lower *K_m* values than *nnLuz_wt* (Supplementary Figure 6).

(6) From the transient assay of *NpgA* in plant cells (Extended Fig. 2), the inclusion of *NpgA* seemed to have the greatest effect (up to 280-fold increase in petunia) on the luminescence intensity among all of the modifications presented in this study. In stable transgenic plants, 8- and 54-fold increase was reported for *N. benthamiana* and *N. tabacum*, respectively. I feel that this was the largest contributor that enhanced luminescence and is worth discussing about the possible reasons. For example, was there a difference in transgene expression level? FBPs protein activity? Endogenous caffeic acid level?

Yes, we believe that *NpgA* was the largest contributor to enhancement of brightness in species lacking endogenous PPTase activity towards HispS (such as petunia). Depending on the endogenous PPTase activity, the relative change in luminescence intensity is different in different systems. We now mention different levels of endogenous PPTase activity in the main text of the revised manuscript.

It is interesting to note that little is known about plant PPTases and their specificity towards recombinant proteins. We found almost no information about plant PPTase activity and substrate specificity towards exogenous proteins.

We are currently working on characterisation of different PPTases of plant origin, and will be soon preparing the publication.

(7) When comparing the different FBP systems (FBP1,FBP2, FBP3), can the transgene expression level of each gene influence the luminescence intensity? Can the transgene expression level change depend on the order of the transgenes within the multiple expression cassette (e.g. possible transgene silencing triggered depending on the position of the transgene in the cassette)? Perhaps, a RT-qPCR to check whether there are any differences in transgene expression level of the different constructs may help. Or confirming the expression of each protein may also help.

We agree with the reviewer's point. Indeed, changes in expression levels of bottleneck genes do translate into intensity of luminescence, as recently shown in detail for the fungal bioluminescence pathway (Calvache et al. 2023), and the order of transcription units does affect expression levels *in planta* (Kallam et al.).

For this revision, we performed additional experiments, which we outline in detail in the response to the point 4 of Reviewer #1 and on **Supplementary Figures 10**. Briefly, changes in the order of transcription units did affect luminescence up to 1.8-fold, however, the major contribution to the overall performance boost was that of improved enzymes. Same conclusions can be obtained from experiments in systems where we used co-transfection of plasmids encoding individual transcription units (BY-2 cells, mammalian cells).

(8) FBP3 was not brighter than FBP2 in stable transgenic *N. benthamiana* lines, which is different to the BY-2, mammalian, yeast cells. Is there any reason for this?

A trivial reason could be that for *N. benthamiana* we only had a single line expressing FBP3 at the time of submission (colour of data points on **Figure 2** indicates individual plant lines). This line could have had integration into a less active genomic locus, resulting in lower expression of genes. For this revision, we created new *N. benthamiana* lines and provided new data on the **Extended Data Figure 5b**. In this case, FBP3 did show brighter luminescence intensity. However, the comparison was performed against a single FBP2-expressing line, for unfortunate technical reasons.

For this revision, we also added new data on performance in *Arabidopsis thaliana* (**Supplementary Figure 20**), which shows the difference in performance between FBP2 and FBP3 that is consistent with data in BY-2, mammalian, and yeast cells.

Our overall recommendation remains to use FBP3 over FBP2 for experiments in plants.

(9) When comparing FBP1 to FBP2 or FBP3 in luminescent plants (**Figure 2**, **Extended data Fig.6**, **Supplementary Fig. 12**, **Supplementary Fig. 14-15**, **Supplementary Fig. 17-18**), which generation of the plant was used? And if different generations were compared, was the luminescence intensity stable over generations?

Thank you for pointing that out, we now provide this information in the revised version of the manuscript in the **Methods** section. Generally, T0 plants were used for experiments. For *Arabidopsis*, T1 plants were used for **Figure 2d**, and T2 - for **Figure 2e**.

We noticed no decrease in luminescence or phenotypic instability over generations – but we did not quantitatively compare luminescence over generations.

(10) nnLuz v4 did not show any enhancement of luminescence intensity in yeast (Supplementary Fig. 8). Is there any reason for this, and I feel it would help the readers if this was discussed.

Thank you for pointing that out, this result was unexpected to us too. For this revision, we repeated transformations to obtain more biological replicates (yeast strains with independent genome integrations), and obtained somewhat different results: nnLuz_v4 did outperform nnLuz_wt.

We updated Supplementary Figure 9 and the corresponding text in the manuscript to reflect these results.

(11) For the HEK293 and pichia assay comparing FBP1 and FBP3 (Supplementary Fig. 10 and 11), was the FBP1 co-transfected with NpgA and FBP3 in a single vector? If this is the case, could the co-transfection efficiency affect the luminescence intensity in these data? and the data would be inconclusive about how much NpgA or H3H_v2 contributed to the luminescence intensity?

We do share the concerns about possible influence of co-transfection efficiency, and tried to take them into account when planning the experiments. In yeast, each gene was transformed consecutively via individual plasmid, and was integrated into the genome, so the reviewer's concern does not apply. In mammalian cells, we provided each gene of the pathway on a separate plasmid. There might have been a difference in co-transfection efficiency due to co-transfection of sets of four vs. five plasmids. But because FBP1 effectively does not glow without NpgA in mammalian cells, and the number of plasmids in the NpgA-less set is lower, we believe the difference between FBP1 and FBP1+NpgA is unlikely arising from differences in co-transfection efficiencies. Other comparisons had the same number and concentration of plasmids per transfection.

We now provide information in figure legends on whether genes were encoded on a single plasmid, or on individual plasmids where appropriate.

(12) The effect of NpgA was negligible in *N. benthamiana* transgenic lines (Supplementary Fig.14), but large in *N. tabacum*. Is there any reason for this?

We believe that this difference between plant species arises from different levels of endogenous PPTase activity or PPTase substrate specificity. NpgA-less *N.benthamiana* lines glow reasonably brightly – in contrast to similar lines of closely related *N.tabacum* which have much lower luminescence. We now discuss the effect of NpgA expression and endogenous PPTase activity in the revised version of the manuscript.

Minor points

Page 1: In the sentence, "Low enzymatic activity and limited stability of enzymes at physiologically relevant temperatures 2 resulted in modest light output..." Should this be quantitative? It may be better to briefly describe the limitations of the FBP1 in relation to its application (e.g. imaging) for the readers to understand the benefit of the new system easier.

We agree. We added a sentence discussing low performance of FBP1 in mammalian cells as an illustration.

Page 2: In the sentence "Similarly to nnHisps, mcitHisps was efficiently activated by phosphopantetheinyl transferase NpgA from *Aspergillus nidulans*, which we confirmed to be a necessary component for bioluminescence in most plant species", the FBP system has been tested mainly on dicot species and it may be an

overstatement to say “most plant species”. Otherwise, if it can be explained that most plant species do not have phosphopantetheinyl transferase, it may clarify the point.

We agree with the reviewer. We could not find much data on endogenous PPTase activity and substrate specificity in plants, thus amended the wording to “most tested plant species”.

Page 2: For the sentence “When stably expressed from a genomic copy, the wild-type fungal pathway FBP1 performed well in tobacco species, however, in our hands it did not yield sufficiently bright luminescence in other species.”, depending on the imaging system, and its purpose, dim luminescence can still be useful. I was not able to follow what would be the required brightness for what application.

We agree with the reviewer that dim luminescence can still be useful, depending on the intended experiment. For example, in petunia, the amount of light produced by plants without co-expression with PPTase was so low that we could barely detect it using our equipment. In contrast, in *Nicotiana* species, the luminescence plants produce could indeed be useful for some experiments. We believe these differences between species reflect different levels of endogenous PPTase activity or specificity – however, even in tobacco, the signal-to-noise ratio improves significantly upon co-expression with PPTase.

We now corrected the sentence in the main text to clarify that in some of the species, no light is emitted upon expression of the NpgA-less version of FBP1.

Page 2: For the sentence “Furthermore, the brightest tissues – petunia flower buds – could be recorded on modern smartphone cameras”, I could not find the image data taken using smartphone cameras.

We now provide a video of petunias shot on iPhone 13 Pro as **Supplementary Video 2**. We also added a similar video of glowing yeast cultures as **Supplementary Video 1**.

Tested on benthamiana, petunia BY-2. Maybe overstatement to say most plant species, especially without demonstrating on monocots.

We agree. We corrected the sentence, replacing “most plant species” with “most tested plant species”.

Figure 1 caption: The comparison of FBP2 is lacking in mammalian and yeast cells. Was there any reason for this, and if FBP2 is to be omitted, the sentence should be rephrased to clarify this.

We now performed additional experiments and updated Figure 1 to include comparison with FBP2.

Figure 1e: It would help the readers to see the difference of luminescence intensity between FBP1, FBP2 and FBP3 as an image for comparison of brightness. This would be important as the comparison in the graph has used tobacco BY-2 cells, which may be different to petunia.

Indeed, as the reviewer pointed out, the difference in relative brightness might be changing from one host to the other.

Figure 1 aims to demonstrate performance of different versions of the pathway in three “unicellular” systems, including BY-2. However, Figure 2 focuses on application in stable plants and provides quantitative comparison between versions of the pathway in four species (two tobacco species, poplar and now also arabidopsis).

We do not have a photo of plants of different species in the same frame, but we report photos of plants as Supplementary Figures (Extended Data Figure 6 shows *N. tabacum* plants, Supplementary Figure 14 – *N. benthamiana*, Supplementary Figure 19 – poplar, Supplementary Figure 20 – arabidopsis) along with corresponding camera settings.

Figure 2: Because the ISO range that was used is large, it would help the readers to understand the relative brightness of each plant by adding the details about the imaging conditions for each species (e.g. ISO, distance to the subject (if they were different for each species)).

Thank you – we now appreciate that the difference between the brightness of different species was not clear to the readers. We added information about imaging conditions to the Methods section.

Extended Fig. 3: Please add the fold-difference compared to the wild-type for understanding the contribution of the different HispS and H3H to the luminescence intensity.

We report the fold-change values as requested.

Extended Fig. 7: It would also be nice to show the wild-type (non-transgenic) plants for phenotypic analysis.

We are not sure we correctly understood the concern, as we do show the wild type (non-transgenic) plants on Supplementary Figure 23.

To provide some context, in our previous study (Mitiouchkina et al. Nat Biotech 2020), we assessed the phenotype of transgenic plants and did find minor differences upon expression of the caffeic acid cycle:

Supplementary Note 2. Toxicity of caffeic acid cycle

The overall phenotype of transgenic plants was similar to the wild type plants suggesting that unlike bacterial bioluminescent system 5, high expression of caffeic acid cycle is not toxic and does not impose an obvious burden on plants (Supplementary Figure 3). More detailed analysis revealed minor differences in carotenoid content and plant height, and no difference in leaf shape and size and chlorophyll content. N. tabacum plants of both transgenic and wild-type lines proceeded from the vegetative state of in vitro culture to the flowering stage within the 8th week after the transfer to greenhouse. While we did not collect quantitative data on seed germination, no obvious difference was observed between wild type and transgenic lines. We also noticed that exposure to intense sunlight resulted in more abundant areas of necrosis in older leaves of transgenic plants in comparison to the wild-type plants.

We did not make such detailed measurements in the present study, but we also did not notice phenotypic changes in transgenic plants created in this work.

Supplementary Fig. 4b (right-side of the graph): How was the “Expression” determined? Gene expression or protein expression level?

In this case, we meant *protein* levels, as quantified by HiBit assay. We understand that this was not clear from the original text of the manuscript: we provided this information in the “Consensus mutagenesis” subsection of Methods, while Supplementary Figure 4 showed mutants obtained with random mutagenesis.

We now added this information to the "Activity of nnLuz and nnH3H mutants in mammalian cells" subsection.

Activity of HiBiT-tagged nnH3H variants was assayed in mammalian cells in co-transfection with nnLuz v3. As a proxy for specific activity, we used the ratio of luminescence (HiBiT/fungal signal) when treated with hispidin or fungal luciferin. Activity of each variant was normalised to the expression level of HiBiT, as quantified with HiBiT lytic reagent (Promega N3030).

Supplementary Fig. 5: Was there any reason the Supplementary Fig. 3 used *E. coli* and Supplementary Fig 5. used *Pichia*? Also, if there are no particular reasons, consistent presentation style may be easier for viewing (i.e. line graph or box and whisker plot).

For historical reasons, our first directed evolution efforts were performed in *E. coli*. We then used *Pichia pastoris* as the main microbial system. Specifically, we validated and tested improved variants in *Pichia pastoris*.

The graphs shown on Supplementary Figures 3 and 5 are visually similar but the corresponding experiments are somewhat different. On Supplementary Figure 3, we show the *kinetics* of luminescence decay of two luciferase variants at a *single temperature*. On Supplementary Figure 5, we report overall light output of luciferase variants incubated at *different temperatures*.

We still feel that the results might be best presented in these different formats.

We re-plotted data on Supplementary Figures 3 to improve style consistency with other figures.

Supplementary Fig. 17 and 18: The two figures are somewhat redundant in the sense that the only difference is the age of the plant. Perhaps it may be better to combine the graphs so it is more obvious that the luminescence intensity has gained over development.

Thank you, we combined Supplementary Figures 17 and 18 – became Supplementary Fig 19.

Graphs in general: I felt that the font size relative to the graph was small, and sometimes difficult to read.

We redesigned Figure 1 to improve the readability, and also optimised the size of other graphs.

Decision Letter, first revision:

Dear Karen,

I hope all has been well since we met at Janelia.

Thank you for submitting your revised manuscript "An improved pathway for autonomous bioluminescence imaging in eukaryotes" (N METH-BC51937B). It has now been seen by the original referees and their comments are below. The reviewers find that the paper has improved in revision, and therefore we'll be happy in principle to publish it in Nature Methods, pending minor revisions to satisfy the referees' final requests and to comply with our editorial and formatting guidelines.

In response to the remaining referee comments we ask that you complete the following:

- (1) Address the minor suggestions regarding clarifications or corrections.
- (2) Better discuss how the method will enable biological discovery in plants and fungi.
- (3) Clearly discuss the need for caffeic acid for the system to work.

We do not ask you to make more stable cell lines to compare FBP2 and FBP3 as requested by Reviewer 3. While we understand their point and agree that these experiments would strengthen your conclusions regarding differences in their performance in mammalian cells, we do not think it would change the overall impact of the paper as a whole.

TRANSPARENT PEER REVIEW

Nature Methods offers a transparent peer review option for new original research manuscripts submitted from 17th February 2021. We encourage increased transparency in peer review by publishing the reviewer comments, author rebuttal letters and editorial decision letters if the authors agree. Such peer review material is made available as a supplementary peer review file. Please state in the cover letter 'I wish to participate in transparent peer review' if you want to opt in, or 'I do not wish to participate in transparent peer review' if you don't. Failure to state your preference will result in delays in accepting your manuscript for publication.

Please note: we allow redactions to authors' rebuttal and reviewer comments in the interest of confidentiality. If you are concerned about the release of confidential data, please let us know

specifically what information you would like to have removed. Please note that we cannot incorporate redactions for any other reasons. Reviewer names will be published in the peer review files if the reviewer signed the comments to authors, or if reviewers explicitly agree to release their name. For more information, please refer to our [FAQ page](https://www.nature.com/documents/nr-transparent-peer-review.pdf).

ORCID

Sincerely,
Rita

Rita Strack, Ph.D.
Senior Editor
Nature Methods

Reviewer #1 (Remarks to the Author):

The following points should be addressed before publication:

- In Figure 1b and c, the fold changes between FBP1 and FBP3 should be added.
- Unlike stated in the rebuttal letter, the temperature in Supplementary Figures 1 and 2 is not given and should be added.

All other points have been satisfactorily addressed.

Reviewer #2 (Remarks to the Author):

Shakhowa et al. developed new bioluminescence imaging tools using a genetically engineered fungal bioluminescence pathway. The novelty of constructing a strong light emitting system using a fungal bioluminescence pathway and the efforts of the authors are worthy of praise. On the other hand, the bioluminescence pathway used in this study has already been reported, and methods for increasing luminescence intensity by modifying existing related genes have also been established (e.g. SCIENCE, 359:6378, 935-939, 2018. DOI: 10.1126/science.aaq1067), so there is no methodological novelty or originality to be found in this research. Furthermore, many examples have already been reported for simply acquiring photos and videos that do not require quantitative analysis using non-CCD cameras, such as iPhone and digital cameras, have been reported, including the aforementioned paper. Therefore, this reviewer's comments to the original manuscript was to ask the authors to clarify the superiority of this system by showing an example where imaging is not possible without using this system. Unfortunately, the authors' responses to my comments were not what I expected.

1) This journal focuses on methodology, and if improving performance in identifying versions of the fungal bioluminescent pathway is a methodological superiority of this study, it should be clearly stated.

2) In response to my comments, the authors described the disadvantages of fluorescence imaging. This reviewer is not looking to see how it differs from existing imaging technologies, and also has hope that "improved version of the pathway reported in this manuscript will enable broader applications of autoluminescence across plant and likely fungal species". The results presented in this manuscript, however, do not seem to represent much of advantages in terms of advances in imaging technology. For example, GFP gene-modified plants allow us to image the plant's growth over time without using special equipment such as a fluorescence microscope (e.g. PNAS 99 (6) 4103-4108, <https://doi.org/10.1073/pnas.052484099>). I hoped that the authors' method, which was established by optimizing the fungal bioluminescence pathway, would demonstrate the superiority of the imaging method by showing examples of imaging that cannot be obtained with GFP gene-modified plants, or by showing applications that enables new imaging when combined with other imaging methods.

3) This reviewer was impressed the strong intensity of the luminescence shown in the revised manuscript (fig. S26 and Video S1) obtained by this system. However, it is unclear how much better it is compared to existing methods as it has not been compared to other imaging systems.

Minor comment;

1) Fig. S1, Fig. S2, and Fig. S3 still include "uM". In Figs. S13cd and S13ef, the concentration of caffeic acid are 100 mM and 220 mM, respectively. Are they correct? The concentrations are thought to be considerably higher than the physiological concentration.

2) My comment was for Fig. S9d (Fig. S10d in the revised manuscript). I don't understand why the authors' answer is for Fig. S9b (Fig. S10b in the revised manuscript).

Reviewer #3 (Remarks to the Author):

Response to summary comment:

The authors have improved the manuscript with additional data showing the contribution of NpgA to the luminescence intensity in each mammalian and yeast as hosts. For the sentence "In plants cells, hispidin synthase and hispidin hydroxylase had comparable contributions", Figures 1, 2, Extended Data Figures 2 and 3 support this well, however, as a minor point, it may be better to rephrase this (e.g. plant cell culture or suspension cells) to distinguish with transgenic plants that did not agree with this result (*A. thaliana* (Figure 2d)).

Response to the revision for each point is described below.

- (1) The authors have included additional data sets which clarified the effect of NpgA, and other components mainly in BY-2 cells. The contribution of each component to the luminescence system is clear. As a minor point, I would like to suggest considering consolidating the data (Figure 1d, Extended Data Figure 3) for each component using plant BY-2 cells for ease of understanding the effect of all components in a single figure.
- (2) I appreciate the authors for their explanation. I have no further comment on this point.
- (3) I appreciate the clarification of fold-change in the Figure. I have no further comment on this point.
- (4) I understand that the authors meant to use this data as a rough benchmark of luminescence compared to other systems. The authors have also added a comparison to the bacterial luminescence system. One needs to keep in mind about the limitations of the assay (e.g. catalytic activity), however, FBP3 clearly shows superior luminescence over the lux system. I have no further comment on this point.
- (5) I understand that there are technical difficulties in performing enzymatic analyses. The authors have additional data showing V_{max} and K_m values using yeast lysates, which shows a clear difference between wild-type nnLuz and nnLuz_v4. I have no further comment on this point.
- (6) The authors have additionally mentioned the different levels of endogenous PPTase activity in the main text, however, I was not able to find the information/data and/or reference. My apologies if I missed the revised sentence, but I would appreciate it if I could get an indication of where the information was added in the text.
- (7) The authors have addressed the possible effect of different orders of the genes on the luminescence intensity with additional data. The data showed up to 1.8-fold increase in luminescence dependent on the order of but does not affect the main conclusion of this study. I have no further comment on this point.

(8) I understand the author's explanation that there were limitations on the number of lines that were used to compare FBP2 to FBP3. I am concerned whether the comparison of FBP2 and FBP3 in stable transgenic lines can be concluded with these results as there was only one transgenic line used for the comparison of both experiments (one line of FBP3 for Fig. 2 and one line of FBP2 for Ext. Fig. 5), which produced a different outcome. I feel multiple independent stable transgenic lines are needed for comparing FBP2 and FBP3.

(9) I appreciate the authors for providing further information about the transgenic generations used in this study. I have no further comment on this point.

(10) The authors repeated the experiment with additional replicates and obtained different results, which show increased luminescence using nnLuz_v4 compared to its wild type. I have no further comment on this point.

(11) I appreciate the explanation and additional information in the manuscript. I have no further comments on this point.

(12) As point (6), I would appreciate an indication of where the information was added in the manuscript.

Minor points

My apologies for the comment in Extended Data Fig. 7 about adding the wild-type plants, which was already shown in Supplementary Figure 23 and described in previous work. The authors have revised and responded to all minor comments appropriately. I have no further comments on these points.

Author Rebuttal, first revision:

Reviewer #1 (Remarks to the Author):

The following points should be addressed before publication:

- In Figure 1b and c, the fold changes between FBP1 and FBP3 should be added.

In the version of Figure 1 discussed by the reviewer, Fig 1b and 1c corresponded to the performance of the pathway in mammalian and yeast systems, respectively. In these hosts, expression of FBP1 without NpgA does not result in *any* light emission, thus we feel it would be misleading for the readers to provide fold-change values in these cases.

We recognise that this was not at all clear and we now added an explanation to the legend of Figure 1.

- Unlike stated in the rebuttal letter, the temperature in Supplementary Figures 1 and 2 is not given and should be added.

Thank you, now corrected: we added this information to figure legends.

All other points have been satisfactorily addressed.

Reviewer #2 (Remarks to the Author):

Shakhowa et al. developed new bioluminescence imaging tools using a genetically engineered fungal bioluminescence pathway. The novelty of constructing a strong light emitting system using a fungal bioluminescence pathway and the efforts of the authors are worthy of praise. On the other hand, the bioluminescence pathway used in this study has already been reported, and methods for increasing luminescence intensity by modifying existing related genes have also been established (e.g. SCIENCE, 359:6378, 935-939, 2018. DOI: 10.1126/science.aag1067), so there is no methodological novelty or originality to be found in this research. Furthermore, many examples have already been reported for simply acquiring photos and videos that do not require quantitative analysis using non-CCD cameras, such as iPhone and digital cameras, have been reported, including the aforementioned paper.

Therefore, this reviewer's comments to the original manuscript was to ask the authors to clarify the superiority of this system by showing an example where imaging is not possible without using this system. Unfortunately, the authors' responses to my comments were not what I expected.

1) This journal focuses on methodology, and if improving performance in identifying versions of the fungal bioluminescent pathway is a methodological superiority of this study, it should be clearly stated.

We updated the text in the discussion with the aim to address how FBP2 and FBP3 enables biological discovery in plants and fungi.

2) In response to my comments, the authors described the disadvantages of fluorescence imaging. This reviewer is not looking to see how it differs from existing imaging technologies, and also has hope that "improved version of the pathway reported in this manuscript will enable broader applications of

autoluminescence across plant and likely fungal species". The results presented in this manuscript, however, do not seem to represent much of advantages in terms of advances in imaging technology. For example, GFP gene-modified plants allow us to image the plant's growth over time without using special equipment such as a fluorescence microscope (e.g. PNAS 99 (6) 4103-4108, <https://doi.org/10.1073/pnas.052484099>). I hoped that the authors' method, which was established by optimizing the fungal bioluminescence pathway, would demonstrate the superiority of the imaging method by showing examples of imaging that cannot be obtained with GFP gene-modified plants, or by showing applications that enables new imaging when combined with other imaging methods.

For plants and other organisms producing a significant amount of autofluorescent compounds, autofluorescence is the biggest issue for non-invasive physiology imaging. The reviewer is correct that fluorescent proteins and fluorescent-protein-based sensors have been extensively applied in plant biology, however, as Moreno and colleagues put it in Handbook of Biological Confocal Microscopy, 2006, "this autofluorescence is a nightmare for many fluorescence applications".

We believe that the improved fungal bioluminescence pathway variants reporter in our paper overcome this problem by combining (1) the ability to fully genetically encode the reporter tool, (2) lack of background signal, (3) high performance across different species of plants and fungi – something that the previous version of the pathway could not provide.

3) This reviewer was impressed the strong intensity of the luminescence shown in the revised manuscript (fig. S26 and Video S1) obtained by this system. However, it is unclear how much better it is compared to existing methods as it has not been compared to other imaging systems.

In the revised version of the manuscript, we provided comparison to the only other autoluminescence pathway known – the bacterial pathway (Extended Data Figure 9 and 10). In plant cells, the performance of the bacterial pathway was orders of magnitude worse than that of the fungal pathway. In mammalian cells, the bacterial pathway outperformed the fungal bioluminescence pathway in our hands.

Minor comment;

1) Fig. S1, Fig. S2, and Fig. S3 still include "uM". In Figs. S13cd and S13ef, the concentration of caffeic acid are 100 mM and 220 mM, respectively. Are they correct? The concentrations are thought to be considerably higher than the physiological concentration.

Thank you for pointing that out. We now replaced "uM" with "µM" in Fig. S1, Fig. S2, and Fig. S3.

The reviewer is correct that caffeic acid concentrations of 100 mM and 220 mM are not physiologically relevant. These were the concentrations we used in some of our experiments. To validate the results at physiologically relevant concentrations, we later used much lower concentrations, such as 100 µM (for example, on Extended Data Figure 1, 12, 13, and others).

2) My comment was for Fig. S9d (Fig. S10d in the revised manuscript). I don't understand why the authors' answer is for Fig. S9b (Fig. S10b in the revised manuscript).

2) Fig. S9d does not seem to reflect the result of Fig. S9c (all combinations are $p < 0.001$). Is Fig. S9d correct?

The current Supplementary Figure S10b combines Supplementary Figure S9c and S9d from the previous version of the manuscript. The exact p-values obtained from the Conover test are listed in the table below. All p-values are below 0.001, as displayed in the sign plot.

In the Supplementary Figure S10d of the current version of manuscript, exact p-values of pairwise Mann-Whitney U-test are 0.001147 and 2.535264e-07 for FBP2 vs nnHisps + nnH3H v2 + nnCPH + NpgA and FBP3 vs mcitHisps + nnH3H v2 + nnCPH + NpgA pairs, respectively.

	FBP1	nnHisps + nnH3H v2 + nnLuz v4 + nnCPH	FBP2	FBP3
FBP1	1	7.928092e-04	8.839769e-15	1.581381e-22
nnHisps + nnH3H v2 + nnLuz v4 + nnCPH	7.928092e-04	1	1.811224e-09	2.476002e-18
FBP2	8.839769e-15	1.811224e-09	1	1.623081e-07
FBP3	1.581381e-22	2.476002e-18	1.623081e-07	1

Reviewer #3 (Remarks to the Author):

Response to summary comment:

The authors have improved the manuscript with additional data showing the contribution of NpgA to the luminescence intensity in each mammalian and yeast as hosts. For the sentence "In plants cells, hispidin synthase and hispidin hydroxylase had comparable contributions", Figures 1, 2, Extended Data Figures 2 and 3 support this well, however, as a minor point, it may be better to rephrase this (e.g. plant cell culture or suspension cells) to distinguish with transgenic plants that did not agree with this result (*A. thaliana* (Figure 2d)).

Thank you, now rephrased.

Response to the revision for each point is described below.

(1) The authors have included additional data sets which clarified the effect of NpgA, and other components mainly in BY-2 cells. The contribution of each component to the luminescence system is clear. As a minor point, I would like to suggest considering consolidating the data (Figure 1d, Extended Data Figure 3) for each component using plant BY-2 cells for ease of understanding the effect of all components in a single figure.

Thank you for the suggestion. We had also considered this, but we felt that the current version of Figure 1 was a good compromise between consolidation of all data in one figure and clarity of communication. We think that for readers, the clarity of the manuscript would suffer if we combined them all in one figure. And in the current form, Extended Data Figure 3 seems straightforward to interpret.

(2) I appreciate the authors for their explanation. I have no further comment on this point.

(3) I appreciate the clarification of fold-change in the Figure. I have no further comment on this point.

(4) I understand that the authors meant to use this data as a rough benchmark of luminescence compared to other systems. The authors have also added a comparison to the bacterial luminescence system. One needs to keep in mind about the limitations of the assay (e.g. catalytic activity), however, FBP3 clearly shows superior luminescence over the lux system. I have no further comment on this point.

(5) I understand that there are technical difficulties in performing enzymatic analyses. The authors have additional data showing V_{max} and K_m values using yeast lysates, which shows a clear difference between wild-type nnLuz and nnLuz_v4. I have no further comment on this point.

(6) The authors have additionally mentioned the different levels of endogenous PPTase activity in the main text, however, I was not able to find the information/data and/or reference. My apologies if I missed the revised sentence, but I would appreciate it if I could get an indication of where the information was added in the text.

The reviewer is correct that no reference or data on different levels of PPTase activity was provided in the manuscript. We *interpreted* different levels of light emission by the PPTase-less version of the pathway as

different levels of endogenous PPTase activity. We agree this was not at all reflected in our discussion. We now corrected the text to highlight that this is just our interpretation.

(7) The authors have addressed the possible effect of different orders of the genes on the luminescence intensity with additional data. The data showed up to 1.8-fold increase in luminescence dependent on the order of but does not affect the main conclusion of this study. I have no further comment on this point.

(8) I understand the author's explanation that there were limitations on the number of lines that were used to compare FBP2 to FBP3. I am concerned whether the comparison of FBP2 and FBP3 in stable transgenic lines can be concluded with these results as there was only one transgenic line used for the comparison of both experiments (one line of FBP3 for Fig. 2 and one line of FBP2 for Ext. Fig. 5), which produced a different outcome. I feel multiple independent stable transgenic lines are needed for comparing FBP2 and FBP3.

We agree that for *N.benthamiana*, a comparison of more stable lines would be ideal. However, we think that based on data in other plants, it's clear that FBP3 generally outperforms FBP2 and should be the pathway variant of choice.

(9) I appreciate the authors for providing further information about the transgenic generations used in this study. I have no further comment on this point.

(10) The authors repeated the experiment with additional replicates and obtained different results, which show increased luminescence using nnLuz_v4 compared to its wild type. I have no further comment on this point.

(11) I appreciate the explanation and additional information in the manuscript. I have no further comments on this point.

(12) As point (6), I would appreciate an indication of where the information was added in the manuscript.

Please see our response to point 6. We now also mention differences in *N.benthamiana* / *N.tabacum* brightness and possible link to PPTase activity in the legend of Supplementary Figure 14.

Minor points

My apologies for the comment in Extended Data Fig. 7 about adding the wild-type plants, which was already shown in Supplementary Figure 23 and described in previous work. The authors have revised and responded to all minor comments appropriately. I have no further comments on these points.

Final Decision Letter:

Dear Karen,

I am pleased to inform you that your Brief Communication, "An improved pathway for autonomous bioluminescence imaging in eukaryotes", has now been accepted for publication in Nature Methods. The received and accepted dates will be March 17, 2023 and Dec 13, 2023. This note is intended to let you know what to expect from us over the next month or so, and to let you know where to address any further questions.

Over the next few weeks, your paper will be copyedited to ensure that it conforms to Nature Methods style. Once your paper is typeset, you will receive an email with a link to choose the appropriate publishing options for your paper and our Author Services team will be in touch regarding any additional information that may be required.

You will receive a link to your electronic proof via email with a request to make any corrections within 48 hours. If, when you receive your proof, you cannot meet this deadline, please inform us at rjsproduction@springernature.com immediately.

Your paper will now be copyedited to ensure that it conforms to Nature Methods style. Once proofs are generated, they will be sent to you electronically and you will be asked to send a corrected version within 24 hours. It is extremely important that you let us know now whether you will be difficult to contact over the next month. If this is the case, we ask that you send us the contact information (email, phone and fax) of someone who will be able to check the proofs and deal with any last-minute problems.

If, when you receive your proof, you cannot meet the deadline, please inform us at rjsproduction@springernature.com immediately.

Once your manuscript is typeset and you have completed the appropriate grant of rights, you will receive a link to your electronic proof via email with a request to make any corrections within 48 hours. If, when you receive your proof, you cannot meet this deadline, please inform us at rjsproduction@springernature.com immediately.

Once your paper has been scheduled for online publication, the Nature press office will be in touch to confirm the details.

Content is published online weekly on Mondays and Thursdays, and the embargo is set at 16:00 London time (GMT)/11:00 am US Eastern time (EST) on the day of publication. If you need to know the exact publication date or when the news embargo will be lifted, please contact our press office after you have submitted your proof corrections. Now is the time to inform your Public Relations or Press Office about your paper, as they might be interested in promoting its publication. This will allow them time to prepare an accurate and satisfactory press release. Include your manuscript tracking number NMETH-BC51937C and the name of the journal, which they will need when they contact our office.

About one week before your paper is published online, we shall be distributing a press release to news organizations worldwide, which may include details of your work. We are happy for your institution or funding agency to prepare its own press release, but it must mention the embargo date and Nature Methods. Our Press Office will contact you closer to the time of publication, but if you or your Press Office have any inquiries in the meantime, please contact press@nature.com.

If you are active on Twitter, please e-mail me your and your coauthors' Twitter handles so that we may tag you when the paper is published.

Please note that *Nature Methods* is a Transformative Journal (TJ). Authors may publish their research with us through the traditional subscription access route or make their paper immediately open access through payment of an article-processing charge (APC). Authors will not be required to make a final decision about access to their article until it has been accepted. [Find out more about Transformative Journals](https://www.springernature.com/gp/open-research/transformative-journals)

To assist our authors in disseminating their research to the broader community, our SharedIt initiative provides you with a unique shareable link that will allow anyone (with or without a subscription) to read the published article. Recipients of the link with a subscription will also be able to download and print the PDF. As soon as your article is published, you will receive an automated email with your shareable link.

Please note that you and your coauthors may order reprints and single copies of the issue containing your article through Springer Nature Limited's reprint website, which is located at <http://www.nature.com/reprints/author-reprints.html>. If there are any questions about reprints please send an email to author-reprints@nature.com and someone will assist you.

Best regards,
Rita

Rita Strack, Ph.D.
Senior Editor
Nature Methods